# HyPoradise: An Open Baseline for Generative Speech Recognition with Large Language Models

**Chen Chen**[1,†]  **Yuchen Hu**[1,†]  **Chao-Han Huck Yang**[2*,3]
**Sabato Marco Siniscalchi**[2,4]  **Pin-Yu Chen**[5]  **Eng Siong Chng**[1]
[1]Nanyang Technological University   [2]Georgia Institute of Technology   [3]NVIDIA Research
[4]Norwegian University of Science and Technology   [5]IBM Research AI
{chen1436,yuchen005}@e.ntu.edu.sg  huckiyang@gatech.edu

## Abstract

Advancements in deep neural networks have allowed automatic speech recognition (ASR) systems to attain human parity on several publicly available clean speech datasets. However, even state-of-the-art ASR systems experience performance degradation when confronted with adverse conditions, as a well-trained acoustic model is sensitive to variations in the speech domain, e.g., background noise. Intuitively, humans address this issue by relying on their linguistic knowledge: the meaning of ambiguous spoken terms is usually inferred from contextual cues thereby reducing the dependency on the auditory system. Inspired by this observation, we introduce the first open-source benchmark to utilize external large language models (LLMs) for ASR error correction, where N-best decoding hypotheses provide informative elements for true transcription prediction. This approach is a paradigm shift from the traditional language model rescoring strategy that can only select one candidate hypothesis as the output transcription. The proposed benchmark contains a novel dataset, "HyPoradise" (HP), encompassing more than 334,000 pairs of N-best hypotheses and corresponding accurate transcriptions across prevalent speech domains. Given this dataset, we examine three types of error correction techniques based on LLMs with varying amounts of labeled hypotheses-transcription pairs, which gains a significant word error rate (WER) reduction. Experimental evidence demonstrates the proposed technique achieves a breakthrough by surpassing the upper bound of traditional re-ranking based methods. More surprisingly, LLM with reasonable prompt and its generative capability can even correct those tokens that are missing in N-best list. We make our results publicly accessible for reproducible pipelines with released pre-trained models, thus providing a new evaluation paradigm for ASR error correction with LLMs.

## 1 Introduction

Automatic speech recognition (ASR) has become increasingly important in modern society, as it enables efficient and accurate transcription of spoken languages. This capability facilitates access to information and enhances communication across various domains, including education [7], healthcare [50], and business [36]. Driven by the recent advances in deep learning, remarkable success has been achieved on several ASR tasks through end-to-end training techniques [28, 27, 9, 22, 30, 100, 15]. However, a major challenge of applying ASR in practical conditions lies in effectively handling variations in speech caused by different factors such as background noise [11], speaker accent [85], and speaking styles [82, 2]. These adverse factors are common and inevitable in speech signal, significantly affecting the accuracy of the recognition results [55].

---

*Work done & open source while the author was at Georgia Tech; Corresponding author. †Equal contribution.

37th Conference on Neural Information Processing Systems (NeurIPS 2023) Track on Datasets and Benchmarks.

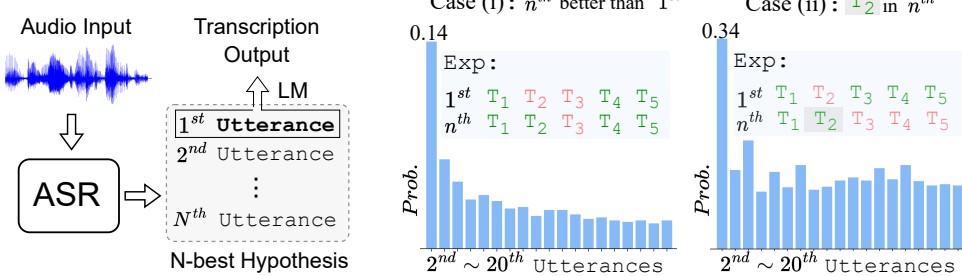

Figure 1: The left part shows the pipeline to generate the N-best hypotheses using a vanilla ASR engine with beam search decoding. The right part counts the probabilities of case (i) and case (ii) on the test set of LibriSpeech dataset. It indicates the discarded information in $2^{nd} \sim 20^{th}$ utterances. Green and red $T_i$ in "Exp" respectively denote correct and wrong tokens compared with ground-truth.

Humans demonstrate remarkable robustness when faced with the above variations in acoustic environment, as the human recognition system does not only rely on acoustic cues – we usually speculate the ambiguous or distorted spoken terms based on speech context and our inherent linguistic knowledge. Similarly, current ASR system typically employs an independent language model (LM) for rescoring during the decoding process [83, 46, 43, 25]. As shown in Fig. 1, given N-best hypotheses generated by an ASR engine with beam search decoding, a trained language model (LM) can be used to re-score each utterance and select the one with the highest likelihood (referred to as the $1^{st}$ utterance) as the output of the ASR; whereas, the other sentences (the $2^{nd} - N^{th}$ utterances) are discarded. However, it is widely believed [68] that the N-best list contains useful information [87, 37, 56], as each hypothesis is an independent textual representation of the input speech. Consequently, discarded sentences might also carry correct tokens for accurately predicting the true transcription. To validate this belief, we have conducted experiments on the LibriSpeech dataset [66], counting the probabilities of two scenarios observed during LM rescoring: (i) the discarded utterances contain a better candidate with lower word error rate (WER), and (ii) the other discarded hypotheses can provide the right answer for the wrong tokens in $1^{st}$ utterance. The statistical results of $2^{nd} \sim 20^{th}$ utterances are shown in the left part of Fig. 1. Taking $2^{nd}$ discarded utterance as example, it has a 14% probability of having a lower WER than the $1^{st}$ utterance. Furthermore, given a wrong token in $1^{st}$ utterance, there is a 34% probability of finding the correct token in the $2^{nd}$ utterance.

To better mine the information in N-best hypotheses, we propose the first attempt on publicly available **ASR generative error correction benchmark** that directly predicts a true transcription, rather than selecting a candidate from the N-best list. To put forth this benchmark, we introduce a novel dataset named ***HyPoradise (HP)***, which comprises various open source N-best hypotheses provided by state-of-the-art ASR systems and their paired true transcriptions. Considering real-life applications, HP dataset covers various challenging speech domains, including scenarios with background noise, specific contexts, and speaker accents. Furthermore, in terms of resources availability, we define three settings to mimic the deployment of ASR systems in real-world scenarios: *(i) Zero-shot* Learning. In this setting, only test set hypotheses are available for inference. This corresponds to applying a well-trained ASR model to new scenarios without any training data. *(ii) Few-shot* Learning. A few in-domain hypotheses with true transcription are available for training. This setting aims to address domain-specific ASR tasks with a few manual annotations. *(iii) Fine-tuning*. A sufficient training set is available to learn the mapping between hypotheses and transcription.

To exploit the three aforementioned scenarios, we present multiple error correction techniques using large language models (LLMs), which have shown the outperforming ability of language generation and reasoning in recent studies [5, 107, 48, 84]. For *zero-shot* and *few-shot* settings, we design an in-context learning method without any parameter tuning, which directly performs error correction based on task prompt and in-domain demonstration. In the *fine-tuning* scenario, we develop two sequence-to-sequence training solutions, H2T-*ft* and H2T-*LoRA*, which adapt pre-trained LLMs to specific transcription domains. Experimental results show that all learning strategies can be beneficial to reduce the WER in different resource settings, providing potential solutions for alleviating the

negative impact of speech variation. Additionally, with reasonable prompt design, LLMs can correct those specious tokens that are exclusive from N-best list. We will release the HP datasets, reproducible pipelines, and pre-trained models on Github [2] under MIT licence.

Our contribution can be summarized as follows:

- We propose the first open and reproducible benchmark to evaluate how LLMs can be utilized to enhance ASR results with N-best hypotheses, where a new dataset HyPoradise [3] with more than 334K hypotheses-transcription pairs are collected from the various ASR corpus in most common speech domains.

- We develop three ASR error correction techniques based on LLMs in different resource settings to directly predict the true transcription from the N-best hypotheses. Experimental results in the *fine-tuning* setting show that our new approach can **surpass** a performance upper-bound (e.g., oracle WER from n-best list) of traditional re-ranking based methods.

- We introduce an evaluation paradigm of *generative error correction* for ASR. The acoustic model generates word-piece elements in the hypotheses list; subsequently, LLMs predict accurate transcription utilizing linguistic knowledge and contextual information.

## 2   Related Work

### 2.1   ASR Rescoring and Error Correction

In order to improve the linguistic acceptability of ASR results, LM rescoring has been widely employed and achieved stable performance gain for ASR systems [79, 62, 4]. Typically, an external LM is trained separately and utilized to re-score the N-best list of hypotheses generated by ASR decoding with beam search. Various approaches for LM integration have been proposed, such as shallow fusion [17, 104, 46, 83], deliberation [98, 32, 41, 40, 91, 39], component fusion [76], and cold fusion [81]. Some authors have used pre-trained LM models to replace trainable LMs [86, 74], and the log-likelihood of each hypothesis is computed using unidirectional models, e.g., GPT-2, or pseudo-log-likelihood using bidirectional models like BERT [21] and RoBERTa [59]. In ASR, LMs are also widely used for the error correction task in different languages [96, 29], leveraging only the 1-best hypothesis generated by the ASR model [53, 61, 106, 23, 109, 77]. Furthermore, more recent works [60, 52, 51] utilize a candidates list after decoding for error correction. Though Grammatical Error Correction (GEC) has been actively explored [20, 93, 100], ASR error correction is distinct with GER due to the arbitrariness of the spoken language [2], which requires the efforts from both speech and NLP communities [18].

### 2.2   Large Language Models

More recently, there has been a surge of interest in Transformer-based LLMs [84, 70, 75, 107] in both academia and industry. By learning from massive amounts of text data, LLMs can capture linguistic patterns and semantic relationships, which have led to impressive performance for a wide range of natural language processing (NLP) tasks [5, 65, 95].

**In-context Learning**. Given specific task descriptions or pair-wise contextual information, LLMs show outstanding adaptability on downstream NLP tasks *without* any parameter tuning [63, 64, 100]. Such a capability of task-specific inference is also known as in-context learning (ICL) [99], which utilize LLMs to generate text that is more coherent and relevant to the specific domain or task [44, 16, 49, 73, 8, 108]. Recently, task-activating Prompting (TAP) [100] is one of the most relevant works, employing the injection of input-output pairs of task-oriented contexts (e.g., initiating the question prompt from a broad domain to refine preceding contexts as shown in Figure 2) with the aim of enhancing the zero-shot and few-shot capabilities of frozen-pretrained LLMs for second-pass ASR. We further evaluate the TAP-based zero-shot and few-shot approaches with examples.

**Low-rank Approximation based Neural Adapter**. Tuning all LLM parameters for a given downstream task is usually not feasible due to memory constraints. Many researchers sought to mitigate that problem by either adapting only a few parameters or leveraging external trainable modules for

---

[2] `https://github.com/Hypotheses-Paradise/Hypo2Trans`
[3] Denoted as *Hypo*theses Pa*radise*, inspired by "Icha Icha Paradise" from Naruto.

a new task [58, 33]. A pioneer work [1] showed that the learned over-parametrized models in fact reside on a low intrinsic dimension, consequently, a low-rank adaptation (LoRA) approach [38] was proposed to indirectly tune some dense layers by optimizing rank decomposition matrices of the dense layers. Due to its computational efficiency, LoRA adaptation has been rapidly adopted as a new paradigm for LLMs tuning, which was useful in various downstream tasks [105, 24, 42, 92].

# 3 Hypothesis Generation and Dataset Creation

We introduce the generation process of the HyPoradise dataset in this section. The employed ASR system for N-best hypotheses generation is illustrated in 3.1, and then we introduce the selected speech domain in 3.2. Finally, we provide statistic information and generated HP in 3.2.

## 3.1 ASR System

We employ two state-of-the-art ASR models, namely WavLM [14] and Whisper [69] for N-best hypotheses generation. Besides their remarkable performance and popularity, those models are representative in the deployment of an ASR because: (1) WavLM is a well-trained ASR model on LibriSpeech [66] but suffering from domain mismatch, and (2) Whisper is a universal ASR model but lacking domain specificity. More details about those two ASR models are described below:

**WavLM**: We utilize the ESPnet toolkit [94] along with the pre-trained model from HuggingFace to deploy our WavLM-based ASR system. The WavLM architecture consists of two blocks: the front-end, and the ASR model (433 million parameters in total). The front-end consists of 24 Transformer-based [88] encoder layers and is pre-trained using a combination of LibriLight [45] (60k hours of data), Gigaspeech [12] (10k hours of data), and VoxPopuli [90] (24k hours of data). Front-end features are fed into the ASR back-end for fine-tuning. The back-end consists of 12 Conformer-based [30] encoder layers, and 6 Transformer-based decoder layers. The fine-tuning process is performed on 960-hour LibriSpeech data. Additionally, the WavLM decoding recipe incorporates an external LM rescoring option, where the external LM adopts Transformer architecture with 16 encoder layers and is trained using the text of LibriSpeech 960 hours data and extra LM training data from the web.

**Whisper**: We employ the Whisper-Large model developed by OpenAI to generate hypotheses, without in-domain language model rescoring. The used configuration consists of an encoder-decoder Transformer architecture with 1,550 million parameters, which is trained on 680,000 hours of multilingual-weakly labeled speech data collected from the web.

Leveraging these two pre-trained ASR models, we have employed the beam search algorithm during decoding and generated N-best lists of sentence hypotheses for each input waveform. For both WavLM and Whisper, the default beam size was set to 60. After removing repeatable utterances, we select top-5 utterances with highest probabilities as N-best list, as they have carried sufficient elements to accurately predict transcription. Subsequent experiments confirm this belief by calculating the accurately upper-bound WER using 5-best hypotheses list. To build the HP dataset, we carry out this decoding strategy on multiple popular ASR datasets (please see Section 3.2) and generate paired data consisting of an 5-best hypotheses list and 1 ground-truth transcription. The pre-processing and generation code are also released for integrating new ASR corpus into HP. All the links of relevant resources are presented in Appendix.

## 3.2 Selected Speech Corpora

For corpora selection, our goal is to cover common scenarios of ASR task, e.g., noisy background and speaker accent. Consequently, we collect and modify the following corpora with evident domain characteristics to compose the HP dataset.

**LibriSpeech** [66]: LibriSpeech is a public corpus of read speech from audiobooks, including 1,000 hours of speech data with diverse speakers, genders, and accents. For generating HP training data, we exclude some simple cases from its *train-960* split that show WER result of 0, resulting in 88,200 training utterances. We use the entire *test-clean* and *test-other* splits for HP test data generation.

**CHiME-4** [89]: CHiME-4 is a dataset for far-field speech recognition. It includes real and simulated noisy recordings in four noisy environments, *i.e.*, bus, cafe, pedestrian area, and street junction. We

Table 1: HP dataset statistics in terms of the number of hypotheses-transcription pairs and average utterance length in various domains.

| Source | Domain Category | Training Set | # Pairs | Length | Test Set | # Pairs | Length |
|--------|-----------------|--------------|---------|--------|----------|---------|--------|
| LibriSpeech | Audiobooks | *train-960* | 88,200 | 33.7 | *test-clean* *test-other* | 2,620 2,939 | 20.1 17.8 |
| CHiME4 | Noise | *train* | 8,738 | 17.0 | *test-real* | 1,320 | 16.4 |
| WSJ | Business news | *train-si284* | 37,514 | 17.5 | *dev93* *eval92* | 503 333 | 16.7 17.3 |
| SwitchBoard | Telephone | *train* | 36,539 | 11.8 | *eval2000* | 2,000 | 11.8 |
| CommonVoice | Accented English | *train-accent* | 49,758 | 10.5 | *test-accent* | 2,000 | 10.5 |
| Tedlium-3 | TED talk | *train* | 47,500 | 12.6 | *test* | 2,500 | 12.6 |
| LRS2 | BBC audio | *train* | 42,940 | 7.6 | *test* | 2,259 | 7.6 |
| ATIS | Airline info. | *train* | 3,964 | 12.4 | *test* | 809 | 11.3 |
| CORAAL | Interview | *train* | 1,728 | 24.2 | *test* | 100 | 24.0 |
| Total | | *train* | 316,881 | 18.1 | *test* | 17,383 | 14.1 |

use its *train* (with 8,738 utterances) and *test-real* (with 1,320 utterances) splits to generate HP training and test data. The four different noises in *test-real* split are also evaluated separately in Table 3.

**WSJ** [67]: The Wall Street Journal (WSJ) is a widely-used benchmark for speech recognition. It includes read speech from speakers in a controlled environment, with a focus on business news and financial data. We use its *train-si284* split (with 37,514 utterances) to generate HP training set. The *dev93* (with 503 utterances) and *eval92* (with 333 utterances) are applied to build test sets.

**SwitchBoard** [26]: The SwitchBoard corpus is a telephone speech dataset collected from conversations between pairs of speakers. It focuses on North American English and involves over 2.4k conversations from approximately 200 speakers. We randomly select 36,539 samples from its *train* split to generate HP training set, as well as 2,000 utterances from the *eval2000* split for HP test set.

**CommonVoice** [3]: CommonVoice 5.1 is a freely-available dataset for speech recognition. It contains speech recordings from diverse speakers in over 60 languages. To generate HP dataset, we randomly select 51,758 samples from its *train-en* split with accent labels, *i.e.*, African, Australian, Indian, and Singaporean, where training set contains 49,758 samples and test set contains 2,000 samples.

**Tedlium-3** [35]: Tedlium-3 is a dataset of speech recorded from TED Talks in multiple languages. It contains a diverse range of background noise, speaker accents, speech topics, etc. Considering its large size, we randomly select 50,000 samples from its *train* split for HP dataset generation, where training set contains 47,500 samples and test set contains 2,500 samples.

**LRS2** [19]: Lip Reading Sentences 2 (LRS2) is a large-scale publicly available labeled audio-visual dataset, consisting of 224 hours of video clips from BBC programs. We randomly select 42,940 samples from its *train* split as training set, and the remaining 2,259 samples are used for test set.

**ATIS** [34]: Airline Travel Information System (ATIS) is a dataset comprising spoken queries for air travel information, such as flight times, prices, and availability. It contains around 5,000 to 5,400 utterances, which are recorded from around 500 to 550 speakers.

**CORAAL** [47]: The Corpus of Regional African American Language (CORAAL) is the first public corpus of AAL data. It includes audio recordings along with the time-aligned orthographic transcription from over 150 sociolinguistic interviews. To generate HP dataset, we select 1,728 samples as training set and 100 samples as test set.

### 3.3 HyPoradise (HP) Dataset Statistics

After performing beam search decoding on the selected speech datasets introduced in Section 3.2, we collected more than 334K pairs of hypotheses list and transcription to form the HP dataset, including training and test sets. The statistics for the HP dataset are given in Table 1, which shows the number

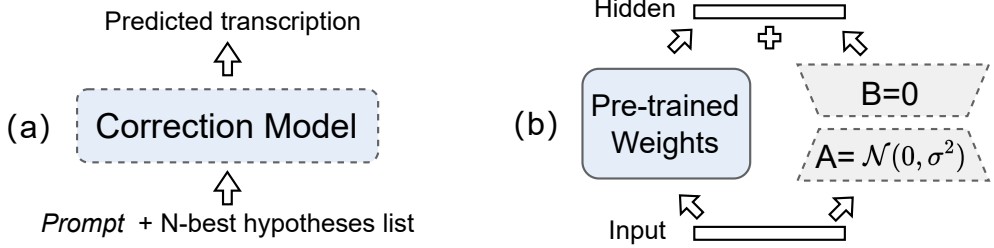

Self-activated In-context Learning

I provide #n real cases in [CLS] domain:

The N-best hypothesis list is: [...],
and the true transcription is: [...]

n-shots demonstration

```
Q: Do you know automatic speech recognition?
R: Yes, ASR is [...]

Q: Do you know language model rescoring in ASR?
R: Sure, LM rescoring [...]

Q: Can you generate an example with 5-best list?
R: 5-best hypothesis list: [...], and answer is [...]

Q: Please do the same thing on following N-best list:
   [N-best hypothesis]
R: Of course. The true transcription should be: [...]
```

Figure 2: A scalable evaluation of Task-Activating Prompting [100] (TAP) based in-context learning. The demonstration in blue box is drawn from the training set, which is optional for LLMs input.

Figure 3: (a) Structure of H2T-*ft*. (b) Reparametrization in H2T-*LoRA*. Solid box denotes the module is fixed during tuning while dashed box stands for trainable. Blue color denotes the weights has been pre-trained on another dataset.

of pairs and average length in various domains and splits. We would release our generated datasets and kindly call for more hypotheses-transcription pairs toward sustainable community efforts.

## 4 ASR Error Correction from Hypotheses to Transcription

We hereby introduce a hypotheses-to-transcription (H2T) training scheme utilizing the collected HP dataset to enhance ASR performance with LLM integration. With limited labeled data, in-context learning [100] is employed to form task-specific prompts and in-domain demonstrations: Linguistic knowledge in LLM is exploited without parameter tuning. Furthermore, we present two trainable methods fine-tuning (*ft*) and H2T-*LoRA* to learn the hypotheses-to-transcription mapping when a sufficient amount of labeled data is available.

### 4.1 Hypotheses-to-Transcription (H2T) Training

In addition to in-context learning, we introduce two parameter-tunable methods to learn hypotheses-to-transcription mapping in a sequence-to-sequence manner: H2T-*ft* and H2T-*LoRA*.

**H2T-*ft*** denotes fine-tuning all parameters of a neural model with labeled data of each HP domain. Specifically, we introduce a similar method with N-best T5, which utilizes other hypotheses to improve the 1-best hypothesis as shown in Fig. 3. To constrain the decoding space, we add an new item criterion $\mathcal{L}_{ft} = \sum_{i=1}^{N} \alpha_i \log P(x^{(i)}|x, \theta)$, where $x^{(i)}$ is the $i$-th hypothesis in N-best list. This item aims to encourage the correction model to preferentially consider tokens into the N-best hypotheses list, preventing arbitrary modification in huge decoding space. $\alpha_i$ is a hyper-parameter for $i$-th hypothesis that decreases with the order ranked by the acoustic model.

**H2T-*LoRA*** avoids tuning the whole set of parameters of a pre-trained model by inserting a neural module with a small number of extra trainable parameters to approximate the full parameter updates, allowing for efficient learning of the H2T mapping without affecting the pre-trained parameters of the LLM. H2T-*LoRA* introduces trainable low-rank decomposition matrices into LLMs' existing layers, enabling the model to adapt to new data while keeping the original LLMs fixed to retain the previous knowledge. Specifically, LoRA performs a reparameterization of each model layer expressed as a matrix multiplication by injecting low-rank decomposition matrices (Fig.3 (b)). As a result, the

Table 2: WER (%) results of H2T-*ft* and H2T-*LoRA* in *fine-tuning* setting. "$o_{nb}$" and "$o_{cp}$" respectively denote n-best oracle and compositional oracle that are defined in 5.2.

| Test Set | Baseline | $LM_{rank}$ | H2T-*ft* | | H2T-*LoRA* | | Oracle | |
|---|---|---|---|---|---|---|---|---|
| | | | T5 | LLaMA | T5 | LLaMA | $o_{nb}$ | $o_{cp}$ |
| WSJ | 4.5 | 4.3 | 4.0 | 3.8 | $2.7_{-40.0\%}$ | $\mathbf{2.2}_{-51.1\%}$ | 4.1 | 1.2 |
| ATIS | 8.3 | 6.9 | 2.7 | 3.4 | $\mathbf{1.7}_{-79.5\%}$ | $1.9_{-77.1\%}$ | 5.2 | 1.1 |
| CHiME-4 | 11.1 | 11.0 | 7.9 | 8.2 | $7.0_{-36.9\%}$ | $\mathbf{6.6}_{-40.5\%}$ | 9.1 | 2.8 |
| Tedlium-3 | 8.5 | 8.0 | 6.6 | 5.2 | $7.4_{-12.9\%}$ | $\mathbf{4.6}_{-45.9\%}$ | 3.0 | 0.7 |
| CV-*accent* | 14.8 | 16.0 | 12.9 | 15.5 | $11.0_{-25.7\%}$ | $\mathbf{11.0}_{-25.7\%}$ | 11.4 | 7.9 |
| SwitchBoard | 15.7 | 15.4 | 15.9 | 18.4 | $14.9_{-5.1\%}$ | $\mathbf{14.1}_{-10.2\%}$ | 12.6 | 4.2 |
| LRS2 | 10.1 | 9.6 | 9.5 | 10.2 | $\mathbf{6.6}_{-34.7\%}$ | $8.8_{-12.9\%}$ | 6.9 | 2.6 |
| CORAAL | 21.4 | 21.4 | 23.1 | 22.9 | $20.9_{-2.3\%}$ | $\mathbf{19.2}_{-10.3\%}$ | 21.8 | 10.7 |

representations generated by the LLM are not distorted due to task-specific tuning, while the adapter module acquires the capability to predict the true transcription from the N-best hypotheses.

Benefiting from efficient training, we can employ a large-scale language model in the H2T-*LoRA* method, which is expected to understand the task description and capture correlation in the N-best list. Meanwhile, instead of adding an extra training objective in H2T-*ft*, we constrain the decoding space of H2T-*LoRA* by adding requirement in task description.

# 5 Experimental Results

## 5.1 Language Models Configurations

**T5** (0.75B∼3B): T5 family [72] is a set of encoder-decoder models pre-trained on a multi-task mixture of unsupervised and supervised tasks and for which each task is converted into a text-to-text format. T5 works well on a variety of tasks out-of-the-box by prepending a different prefix to the input corresponding to each task, e.g., for machine translation or text summarization. In this paper, we select T5-*large* (0.75B) as the correction model in H2T-*ft* method.

**LLaMA** (7B∼65B): Proposed by Meta AI, LLaMA [84] is a collection of foundation language models ranging from 7B, 13B, 30B, and 65B parameters. It is trained on publicly available datasets exclusively, and shows remarkable efficiency on NLP benchmarks. We select LLaMA-13B for LoRA adaptation in H2T-*LoRA* method as one best setup under ablations.

**GPT-3.5** (175B): Proposed by OpenAI, GPT-3.5-turbo is one of the most advanced large language models, which powers the popular ChatGPT. It has been optimized from the GPT-3 [5] for chat purposes but works well for traditional completions tasks as well. We utilize GPT-3.5-turbo in task-activated in-context learning [100], which conduct *zero-shot* and *few-shot* learning experiments with designed task prompt.

## 5.2 Training and Evaluation

For *few-shot* settings, the specific task prompts, along with the LLM's responses from task-activated ICL prompting [100], are provided in the Appendix (page 20). For *fine-tuning* setting, the detailed configuration of H2T-*ft* and H2T-*LoRA* are also explained in Appendix. Furthermore, we release some of the pre-trained correction models to allow interested readers to reproduce our results.

We report WER results as the evaluation metric for all methods. Additionally, we report the two oracle WER for comparison, which are 1) the n-best oracle $o_{nb}$: WER of the "best candidate" in N-best hypotheses list, and 2) the compositional oracle method $o_{cp}$: achievable WER using "all tokens" in N-best hypotheses list. The $o_{nb}$ can be viewed as upper bound performance of the re-rank based method, while $o_{cp}$ denotes the upper bound of correction using occurred elements in the list.

Table 3: Cross-domain WER results by task-activated ICL [100] in *zero-shot* and *few-shot* settings. "$o_{nb}$" and "$o_{cp}$" respectively denote n-best oracle and compositional oracle that are defined in 5.2.

| Domain Shift | Test Set | Baseline | $n$-shot In-Context Learning (ICL) | | | | Oracle | |
| | | | $n = 0$ | $n = 1$ | $n = 5$ | $n = 10$ | $o_{nb}$ | $o_{cp}$ |
|---|---|---|---|---|---|---|---|---|
| Specific Scenario | WSJ-*dev93* | 9.0 | $8.5_{-5.6\%}$ | $7.8_{-13.3\%}$ | $7.7_{-14.4\%}$ | $7.1_{-21.1\%}$ | 6.5 | 5.3 |
| | WSJ-*eval92* | 7.6 | $7.3_{-3.9\%}$ | $6.6_{-13.2\%}$ | $6.6_{-13.2\%}$ | $6.3_{-17.1\%}$ | 5.5 | 4.7 |
| | ATIS | 5.8 | $5.5_{-5.2\%}$ | $5.1_{-12.1\%}$ | $5.0_{-13.8\%}$ | $4.7_{-19.0\%}$ | 3.5 | 2.4 |
| Common Noise | CHiME4-*bus* | 18.8 | $17.6_{-6.4\%}$ | $16.7_{-11.2\%}$ | $16.2_{-13.8\%}$ | $15.9_{-20.7\%}$ | 16.8 | 10.7 |
| | CHiME4-*caf* | 16.1 | $14.7_{-8.7\%}$ | $14.3_{-11.1\%}$ | $13.7_{-14.9\%}$ | $13.2_{-18.0\%}$ | 13.3 | 9.1 |
| | CHiME4-*ped* | 11.5 | $10.9_{-5.2\%}$ | $9.9_{-14.4\%}$ | $9.7_{-15.7\%}$ | $9.4_{-18.3\%}$ | 8.5 | 5.5 |
| | CHiME4-*str* | 11.4 | $10.9_{-4.4\%}$ | $10.0_{-12.3\%}$ | $9.7_{-14.9\%}$ | $9.2_{-19.3\%}$ | 9.0 | 6.0 |
| Speaker Accent | CV-*af* | 25.3 | $24.9_{-1.6\%}$ | $24.2_{-4.3\%}$ | $23.6_{-6.7\%}$ | $22.6_{-10.7\%}$ | 23.6 | 21.7 |
| | CV-*au* | 25.8 | $25.1_{-2.7\%}$ | $24.1_{-6.6\%}$ | $24.0_{-7.0\%}$ | $23.3_{-9.7\%}$ | 24.9 | 21.8 |
| | CV-*in* | 28.6 | $27.6_{-3.5\%}$ | $25.6_{-10.5\%}$ | $25.0_{-12.6\%}$ | $24.4_{-14.7\%}$ | 27.1 | 22.6 |
| | CV-*sg* | 26.4 | $26.5_{+0.4\%}$ | $25.0_{-5.3\%}$ | $25.1_{-4.9\%}$ | $23.7_{-10.2\%}$ | 25.5 | 22.2 |

Table 4: Case study of ICL. The utterance is drawn from WSJ-*dev93* dataset.

| Type | Utterance | WER |
|---|---|---|
| $1^{st}$ Hypo. by AM | Bankers in Hong Kong expect xinnepec to return for more loans as it develops China's petro chemical industry. | 16.7 |
| $2^{nd}$ Hypo. by AM | Bankers in Hong Kong expect xinepec to return for more loans as it develops China's petrochemical industry. | 8.3 |
| Correction by LLM | Bankers in Hong Kong expect Sinopec to return for more loans as it develops China's petrochemical industry. | 0 |
| Ground-truth Transcription | Bankers in Hong Kong expect **Sinopec** to return for more loans as it develops China's **petrochemical** industry. | - |

## 5.3 Results of H2T-*ft* and H2T-*LoRA*

We first report the WER results for H2T-*ft* and H2T-*LoRA* in the *fine-tuning* setting, where the training set of HP is available to learn H2T mapping. Whisper is employed as acoustic model for hypotheses generation, and a vanilla language model $LM_{rank}$ is trained using in-domain transcription of the training set, and then it re-ranks the hypotheses according to perplexity. From Table 2, we observe that 1) correction techniques achieve significant performance gain in specific scenarios, where H2T-*LoRA* respectively reduces 77.1% and 55.1% relative WER on ATIS and WSJ. 2) WER performances on CHiME-4 and CV-*accent* demonstrate proposed correction methods improves the robustness of on background noise and speaker accent. Additionally, H2T-*LoRA* on these two datasets both surpass the upper-bound of re-ranking based method referring to $o_{nb}$. 3) In general, H2T-*LoRA* usually generate better WER results than H2T-*ft*, as the low-rank adapter allows LLMs to keep pre-trained knowledge and avoid over-fitting problem.

**Limitation and Failure Studies.** We notice that an over-fitting phenomenon existing in our correction techniques, especially in H2T-*ft* where all parameters are tunable. Furthermore, the mean and variance of the utterance length can potentially influence the WER result, since H2T-*ft* results on CORAAL (long-form speech) and SwitchBoard (large variance in length) both fail to enhance ASR performance. On LibriSpeech, when the WER is low (1.8% by WavLM), there is less room to correct recognition errors with proposed framework. The experimental results and list the representative failure cases can be found in Appendix Table 6 and Table 7. Given the evidence of ample room for further performance improvement, our proposal thus serves as an appropriate benchmark to assess the contribution of current and future LLMs to ASR.

## 5.4 In-context Learning Results

We conduct in-context learning experiments in the practical scenario when a well-trained ASR system encounters domain mismatch. To this end, the WavLM is selected as the in-domain acoustic

model, and GPT-3.5 serves as the LLM for correction. We mainly consider common domain shifts of application: specific scenario, common background noise, and speaker accent, where 5-best hypotheses are selected as context input. From Table 3, we can observe that: (1) Without any in-domain data, LLM can benefit from ASR results based on the hypotheses list. This performance gain mainly relies on the linguistic knowledge of LLM and task-activating [100] descriptions (e.g., chains of task hints) in pipeline. (2) A few in-domain pairs effectively enhance the performance gain in terms of WER. From the final output of the reasoning process, we find that LLM attempts to summarize the regulation from the demonstration and then apply it to the given test example. (3) Leveraging the vast knowledge base, LLM can even correct missing tokens that are exclusive from hypotheses list in terms of context information.

To illustrate the third observation, we conduct the case study on WSJ-*dev93* in Table 4. According to the ground-truth transcription, two errors (shown as red) are included in $1^{st}$ hypothesis, where "petro chemical" is wrongly recognized as two tokens perhaps due to the speaking style of the speaker. LLM correct this error since "petrochemical" can be found in $2^{nd}$ hypothesis. However, "Sinopec" is unseen during ASR training, leading it to be recognized as weird tokens ("xinnepec" or "xinepec") in hypotheses. In this case, LLM shows human-like correction – it successfully infers the correct token based on the pronunciation of "xinnepec", as well as the context of "China's petrochemical". In fact, Sinopec is a petrochemical-related Chinese company.

### 5.5 Additional Discussion

**Effect on Spoken Language Intent Detection.** We examine the effect of error correction on a downstream task of spoken intent detection [80] (SID). To this end, we reproduce an BERT-based SID model [13] and respectively feed the 1-best utterance and corrected utterance by H2T-*LoRA* for comparison. The ablation results on ATIS dataset are reported in Appendix, which shows that our correction technique can also benefit to SID task in terms of detection accuracy. (3) LLM correction based on N-best hypotheses can effectively enhance the downstream SIT result, which achieves comparable accuracy with using ground-truth transcription (97.4% *v.s.* 97.9%).

**Zero-shot Prompting Results.** We finally report an initial prompting evaluation on CHiME-4 in *zero-shot* setting. Considering the task difficulty, T5 and LLaMA are employed for hypothesis correction. For comparison, we also provide the correction results using a far smaller GPT-2 (1.5B) with a 5-gram LM baseline trained by in-domain transcription. We used LLaMA 13B to perform these zero-shot error correction tasks. Using the test set extracted from Whisper, we observed that the zero-shot method did not yield improved results on CHiME-4 ($11.5 \pm 0.5\%$) and CV-accent ($14.9\% \pm 1.5\%$). This zero-shot pipeline performed less stably on the other test set discussed in Table 2, which we consider a failure case with a standard deviation exceeding an absolute value of 10% in terms of WER. For T5-based error correction, we noticed that the method also failed to perform zero-shot error correction by using 0.75B.

**Future work**. We find that LLMs potentially perceive acoustic information during pre-training, as they tend to perform error correction using tokens with similar pronunciation. Therefore, our first future work is including more acoustic information in HP dataset, such as token-level confidence provided by ASR engine. Furthermore, considering different data amount of each domain, more parameter-efficient training methods besides low-rank adaptation should be discussed for LLMs tuning [54], e.g., model reprogramming [102, 31], prompting [10] and cross-modal adaptation [97, 101, 71].

## 6 Conclusion

To explore the benefits in speech-language co-learning, this work introduces a new ASR benchmark that utilizes LLMs for transcription prediction from N-best hypotheses. Our benchmark contains a new HP dataset consisting of more than 334K hypotheses-transcription pairs that are collected from 9 different public ASR corpora. In *few-shot* settings, we demonstrate that LLMs with in-context learning can serve as a plug-and-play back end to effectively alleviate domain shift of ASR. In the *fine-tuning* setting, our proposed error correction technique based on LLMs achieves better WER performance than the upper-bound of re-ranking based method, which provides a new paradigm for applying ASR in some challenging conditions, such as background noise and speaker accent. We believe our benchmark and findings provide new and unique insights into LLM-enhanced ASR.

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

# Appendix

## Have LLMs seen ASR transcription during pre-training?

We raise a concern about LLM correction: is the remarkable correction result owing to to the fact that the ASR transcriptions exist in the pre-trained corpus of LLM?

To evaluate this concern for GPT-3.5, we manually create some hypotheses to simulate the utterance in case study in Table 4, and we show the correction results in Table 5.

Table 5: Synthetic hypotheses for LLM correction

| ID | Utterance |
|----|-----------|
| 1 | Patrue China and xinnepec own all national oil pipeline infrastructure. 
 Corrected item: "Patrue China" → "PetroChina", "xinnepec" → "Sinopec" |
| 2 | The Legend of Saida is the most anticipated game of this year from nine ten door. 
 Corrected item: "Saida" → "Zelda", "nine ten door" → "Nintendo" |
| 3 | The world error rate is the primary matrix to evaluate ASR performance. 
 Corrected item: "world" → "word", "matrix" → "metric" |
| 4 | He missed the mid nine flight from Singapore to rose island. 
 Corrected item: "mid nine" → "midnight", "rose" → "Rhodes" |

We observe that given arbitrarily fabricated utterances in different scenarios, LLM can still give reasonable correction results, especially on specific terms. Furthermore, we attach the LLM reasoning process for two examples, which shows the utilization of context information:

*ID 1: Please note that the transcription assumes "Patrue China" was an error, and "PetroChina" was intended instead. Additionally, "xinnepec" has been interpreted as "Sinopec," which is another major Chinese oil and gas company.*

*ID 2: Please note that there may have been some confusion with the phrase "nine ten door" as it doesn't match any known company or widely recognized term. As a result, I have inferred that it may have been a misinterpretation or a typographical error. The transcription assumes "Zelda" as a possible replacement for "Saida" based on its similarity to the well-known game series "The Legend of Zelda" developed by Nintendo.*

In H2T-*ft*, the T5 model was pre-trained on C4 dataset that does not include WSJ, but it still achieves significant WER reduction on WSJ. In H2T-*LoRA*, we do not find the any ASR transcript exists in the pre-trained data of LlaMa according to their papers or public information.

In general, we argue that the improved results of LLMs correction primarily stem from its extensive knowledge base and the ability to utilize context information as part of a *formal scientific experimental process*. However, the methods through which indirect knowledge is captured and propagated across a large area remain open questions for future investigation. We hope that by initiating this study, the results would encourage more further and ongoing works on knowledge grounding in the future.

## Resources for Hypotheses Paradise (HP) Generation

The related toolkit and pre-trained models used to generate HP dataset are listed as follows:

- For ASR model, we used pre-trained WavLM from HuggingFace[4] and the ESPNet toolkit[5].
- When using WavLM for ASR decoding, we also incorporate an external LM rescoring option, which is trained using extra text data from web[6].
- Apart from WavLM, we also select Whisper-LargeV2[7] for ASR decoding.

---

[4] `https://huggingface.co/espnet/simpleoier_librispeech_asr_train_asr_conformer7_wavlm_large_raw_en_bpe5000_sp`

[5] `https://github.com/espnet/espnet/tree/master/egs2/librispeech/asr1`

[6] `http://www.openslr.org/resources/11/librispeech-lm-norm.txt.gz`

[7] `https://github.com/openai/whisper`

## Task-Activated Prompting (TAP) for Error Correction Evaluation

We illustrate the given prompt for task-activated prompting [100] (TAP), as well as the response from the frozen pretrained LLM for each turn, as proposed by Yang et al. [100] in a recent work.

*Q*: *Are you familiar with speech recognition?*

*R*: *Yes, I am familiar with speech recognition. Speech recognition, also known as automatic speech recognition (ASR) or speech-to-text, is the process of converting spoken language into text. This technology involves using algorithms and machine learning models to analyze and transcribe the acoustic features of spoken words and phrases. Speech recognition has many applications, including voice-controlled assistants, automated phone systems, and transcription services.*

*Q*: *Are you familiar with language model rescoring in ASR?*

*R*: *Yes, I am familiar with language model rescoring for speech recognition. Language model rescoring is a technique used to improve the accuracy of speech recognition systems. It involves using a separate language model to evaluate the likelihood of a given hypothese list. This separate model is typically more complex and powerful than the initial language model used for the transcription, and it is used to re-score the transcription based on the probability of the words occurring in the given context. The rescoring process involves taking the output of the initial language model, which is usually based on statistical methods such as Hidden Markov Models, and then applying a more advanced language model, such as a neural network-based language model, to generate a more accurate transcription. This is accomplished by re-ranking the possible transcriptions based on the probabilities assigned by the more advanced language model. Language model rescoring has been shown to significantly improve the accuracy of speech recognition systems, particularly in noisy or challenging environments where the initial language model may not perform well.*

*Q*: *Can you give a possible example on language model rescoring with 5-best hypotheses?*

*R*: *Sure, here is an example of language model rescoring for ASR with 5-best hypotheses:*

> *1. I want to go to the store.*
>
> *2. I want to go to the storm.*
>
> *3. I want to go to the stove.*
>
> *4. I want to go to the star.*
>
> *5. I want to go to the storage.*

> *After rescoring, I think the ground-truth of this speech should be: I want to go to the store.*

*Q*: *Nice job, i will give you a real example as a demonstration from [target domain]. The 5-best hypothesis is:[hypotheses list from training set], and I expect your output is: [corresponding transcription]. Following this example, can you report the true transcription from the following 5-best hypotheses:? [hypotheses list for inference]*

It is noted that the responses from GPT-3.5 vary from random seeds. In our experiments, we utilize the API key from OpenAI, and fix the above template before the last turn error correction.

We observe that many recent works [103] of ICL focus on the selection strategy for in-domain demonstration, which might have an obvious impact on inference results. We leave the demonstration selection as future work, and in our *few-shot* learning, we manually select those utterances with long lengths according to [64].

## Hypotheses-to-Transcription (H2T) Training Configuration

**H2T-*ft***. We employ the T5-v1.1-large pre-trained model (0.75B) downloaded from HuggingFace [8]. Compared with the original T5 model, GELU Sevres as activation function in the feed-forward layer to replace ReLU. Furthermore, T5 Version 1.1 was only pre-trained on C4 excluding any supervised training. Therefore, this model has to be fine-tuned before it is applied on a downstream task.

---

[8] https://huggingface.co/google/t5-v1_1-large

We finetune 20 epochs on each domain of HP dataset with a batch size of 16. To select the best model, we first split a validation set with 5% data amount of training set. The learning rate varies from $1 \times e^{-4} \sim 1 \times e^{-3}$ according to data amount of each domain, and AdamW is employed for optimization. The $\alpha_1$ to $\alpha_1$ are set as 0.1, 0.05, 0.05, 0.05 respectively, as the $2^{nd}$ utterances are usually more informative than others as shown in Fig.1. In practice, we observe the over-fitting phenomenon during training. The WER on training set can be lower than 1%, however, the performance on CORAAL dataset even is even worse than the baseline. In other words, H2T-*ft* still has room for improvement by adding some techniques for avoiding over-fitting.

**H2T-*LoRA*.** We select LlaMa-13B as the frozen pre-trained model in our method, which is downloaded from HuggingFace [9]. The learning rate is set as $1e^{-4}$, and the batch size is 128. For the low-rank adapter, we implement by peft [10], where the configuration of rank $r$ is set as 8. Similarly, we also use T5-v1.1-large as pre-trained model with low-rank adapter for experiments, where the learning rate is set as $3e^{-4}$ and the lora_$r$ is set as 16.

We train 10 epochs using AdamW optimizer, and the prompt for LLM is designed as follows:

*"Below is a best-hypotheses that is transcribed from an automatic speech recognition system. Write a response to predict the true transcription using the tokens from other-hypotheses.### best-hypothesis:{$1^{st}$ utterance}### other-hypothesis:{$2^{nd} \sim 5^{th}$ utterances} ###Response:"*

The prompt template is not unique, and it leaves a slight impact on the final WER result. Additionally, we calculate the WER using Sclite[11] toolkit, which keep consistent with evaluation script of ESPNet[12].

$LM_{rank}$ is an Transformer-based language model that is implemented using ESPNet toolkit [13], where the training transcription from each HP domain is utilized for a typical LM training. The Transformer layer of each model varies from 8 to 16 in terms of data amount. The training epoch is set as 20, and Adam is employed as optimizer. The initial learning rate is set as 0.002 with warm up strategy. During decoding, the perplexity of each hypothesis is calculated for re-ranking the N-best list, and the utterance with the lowest perplexity is selected as the final output.

## LibriSpeech Results and Failure Cases Study

We list two representative failure cases from LibriSpeech-*test-other* in Table. 7. For the first case, "ward" is corrected by "warde" as there is an "his" behind it. Additionally, we observe that "warde" also appears in the $2^{nd}$ hypothesis, so LLM adopts it according to context information. For the second case, LLM directly adopts the $2^{nd}$ utterance in the N-best list, as "think" does not often appear at the beginning of a sentence from a grammatical perspective. Therefore, as [100] and explained in future work of 5.5, we argue that LLM correction should also consider acoustic information provided by the ASR system, which helps to avoid "over-correction" cases and keeps the fidelity to spoken language.

Table 6: WER (%) results on LibriSpeech dataset. "$o_{nb}$" and "$o_{cp}$" respectively denote n-best oracle and compositional oracle that are defined in 5.2.

| Test Set | Baseline | $LM_{rank}$ | Correction *with* H2T-*ft* | H2T-*LoRA* | *Oracle* $o_{nb}$ | $o_{cp}$ |
|---|---|---|---|---|---|---|
| LS-*clean* | 1.8 | 1.8 | $1.8_{-0.0\%}$ | $\mathbf{1.7}_{-5.6\%}$ | 1.0 | 0.6 |
| LS-*other* | 3.7 | 3.7 | $3.9_{+5.4\%}$ | $3.8_{+2.7\%}$ | 2.7 | 1.6 |

## Results on Mandarin and code-switching Dataset

We include AISHELL-1 [6] as a Mandarin dataset into HP benchmark, consisting of a training set with 120098 utterances and a testing set with 7176 utterances. We randomly select 20k examples (16.7%) from training set to evaluate the effect of proposed H2T-*LoRA*. For the foundation model, we

---

[9] https://huggingface.co/decapoda-research/llama-13b-hf

[10] https://github.com/huggingface/peft

[11] https://github.com/usnistgov/SCTK/blob/master/doc/sclite.htm

[12] https://github.com/espnet/espnet/blob/master/egs2/TEMPLATE/asr1/asr.sh

[13] https://github.com/espnet/espnet/tree/master/egs2/librispeech/asr1

Table 7: Failure cases corrected by H2T-*LoRA*. The utterances are drawn from LibriSpeech-test-*other*.

| Type | Utterance | WER |
|---|---|---|
| $1^{st}$ Hypo. by AM | Yet there was gambling again the second night between ward and several others of his profession. | 0 |
| Correction by H2T | Yet there was gambling again the second night between warde and several others of his profession. | 6.25 |
| Ground-truth Transcription | Yet there was gambling again the second night between **ward** and several others of his profession. | - |
| $1^{st}$ Hypo. by AM | Think he really needs it he pursued | 0 |
| Correction by H2T | He really needs it he pursued | 14.3 |
| Ground-truth Transcription | **Think** he really needs it he pursued | - |

employ Chinese LlaMa2-7b from Huggingface[14], and keep other settings consistent with H2T-*LoRA* in this paper. Notably, ASR on Mandarin dataset is usually evaluated by character error rate (CER), as character is equal to word in Chinese. We also evaluate H2T-*LoRA* on ASRU [78] that is a Mandarin-English code-switching dataset. Each utterance in ASRU contains 8.6 Chinese characters and 1.6 English words on average, and the transcripts cover many common fields, including entertainment, travel, daily life, and social interaction. Similarly to AISHELL-1, Chinese LlaMa2-7b is selected as foundation model, and 15k Hypotheses-Transcription pairs are randomly used in H2T training. Additionally, the evaluation metric for code-switching ASR is mixed error rate.

From the results reported in Table 8, we observe that H2T-*LoRA* demonstrates its generalization on both Mandarin and code-switching dataset, which respectively achieves 20.6% CER and 24.5% MER reductions. Furthermore, H2T-*LoRA* exhibits remarkable data efficiency since the training pairs is equivalent to less than 20 hours of speech data.

Table 8: Mandarin and code-switching results on AISHELL-1 (CER) and ASRU (MER) dataset, where 20k and 15k Hypotheses-Transcription pairs are respectively used in H2T training.

| Dataset | Language | Baseline | H2T-*LoRA* | $o_{nb}$ | $o_{cp}$ |
|---|---|---|---|---|---|
| AISHELL-1 | Man | 6.3 | $5.0_{-20.6\%}$ | 4.1 | 3.1 |
| ASRU | Man & Eng | 11.0 | $8.3_{-24.5\%}$ | 8.6 | 6.5 |

**Results on Spoken Language Intent Detection (SID) task**

We first train an intent detection model using the transcription of ATIS training set, as the intent label is available for each example. Then, during testing, we respectively feed the $1^{st} \sim 5^{th}$ utterances in Whisper hypotheses list, utterance after correction, and ground-truth transcription as input text for intent detection. The accuracy results are reported in Table 9.

Table 9: Accuracy (%) results of intent detection with different input on ATIS test set.

| Textual input | $n^{th}$ utterance in Hypotheses list, $n =$ | | | | | After Correction | Oracle |
|---|---|---|---|---|---|---|---|
| | 1 | 2 | 3 | 4 | 5 | | |
| Acc. (%) | 94.9 | 95.5 | 94.2 | 94.3 | 94.2 | $\mathbf{97.4}_{+2.5\%}$ | 97.9 |

---

[14]https://huggingface.co/ziqingyang/chinese-llama-2-7b

We observe that: (1) When we use corrected text for intent detection, the accuracy is 97.4% which achieves an absolute improvement of 2.5% over $1^{st}$ utterance in the hypothesis list. (2) $2^{nd}$ utterance is more suitable for intent detection than $1^{st}$ utterance in terms of accuracy. This phenomenon validates the case (ii) from a perspective other than WER, where the discarded utterances in the N-best hypotheses might be better than the selected utterance.

**Preliminary Results on Zero-shot Prompting-based Error Correction**

To examine the zero-shot ability of LLM, we propose a framework that requires a targeted LLM to perform either (i) ranking-based error correction or (ii) single-sentence generative correction. We follow the self-activated prompting method mentioned in the previous appendix section to repurpose the language model in the form of zero-shot error correction, without providing instructions. Using the same decoding test set from WavLM and Whisper, a 5-gram language model (coefficient of 0.1) combined with its acoustic model score showed a 2.95% WER relative improvement. This result is slightly worse than the $\text{LM}_{rank}$ baseline. The current limitations on the results regarding the zero-shot abilities of LMs could be attributable to the model scale. The zero-shot or emergent abilities of these models have been reported [100] to be more significant when the parameter scale of the LLM exceeds 100B.

**Hypotheses Paradise (HP) Dataset Visualizations**

We have open-sourced a Colab example[15] for HP dataset visualizations and analysis. First, same as Fig. 1, we visualize and analyze the information in N-best hypotheses from both utterance- and token-levels. Fig. 4 illustrate more visualizations on CHiME-4 test sets, where we can observe valuable information in N-best hypotheses.

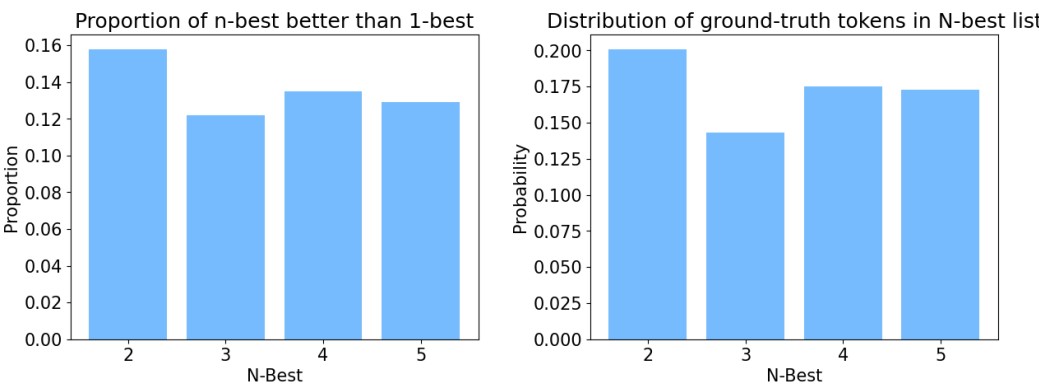

Figure 4: Probabilities of the case (i) and (ii) on CHiME-4 test set, similar to the right part of Fig. 1.

Furthermore, we also visualize and compare the word frequency in N-best hypotheses and ground-truth transcription in Fig. 5, where we can observe some but limited gap between them.

**N-best Hypotheses Distribution**

Fig. 6 visualizes the distribution of N-best hypotheses generated by different-sized Whisper models, *i.e.*, from 'tiny' to 'large'. We can observe very limited diversity in the N-best hypotheses generated by Whisper models. Considering such high monotonicity, we only collect the top-5 hypotheses to form our HP dataset.

## Limitations

Though the proposed HP benchmark provides a new paradigm of generative error correction for ASR, we analyze and discuss the limitations of this work from the following perspective:

---

[15]`https://colab.research.google.com/drive/1traA2scdnmAKFq6yIEZhHwrhCBVxB2ig`

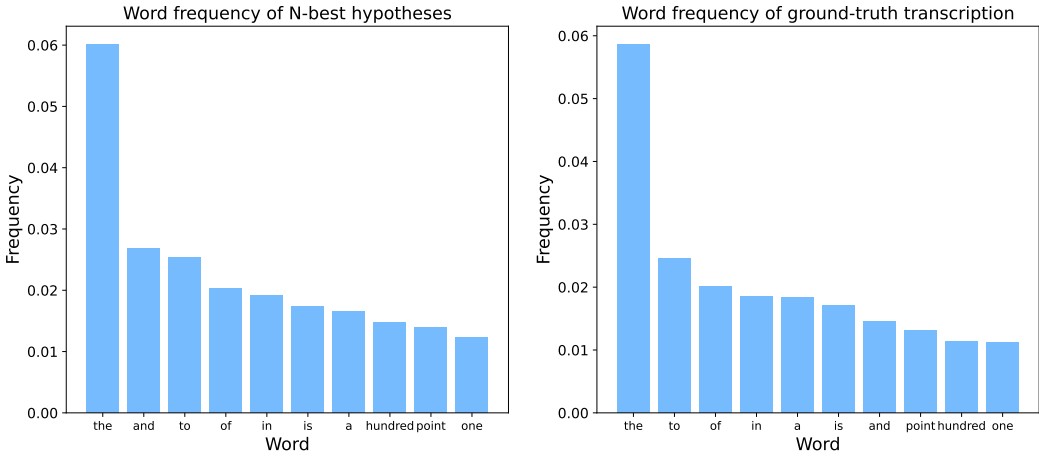

Figure 5: Top-10 word frequencies in N-best hypotheses and ground-truth transcription of CHiME-4 test set.

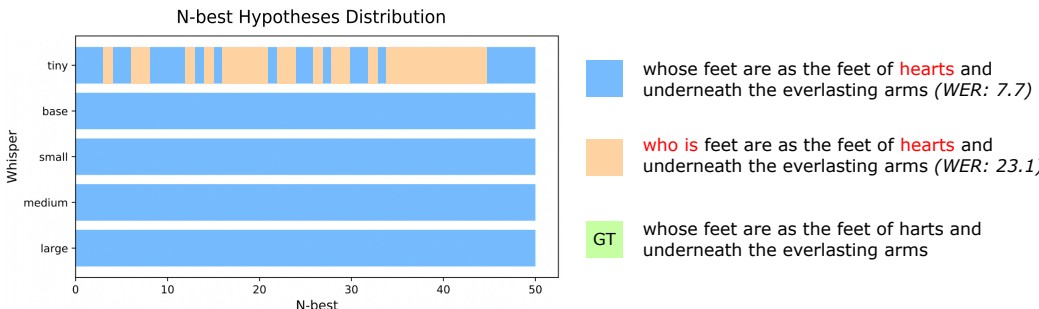

Figure 6: N-best hypotheses distribution with different Whisper models. Each color denotes an unique hypothesis, 'GT' denotes the ground-truth transcription. The sample is selected from LibriSpeech test-clean set, *i.e.*, '1089/134691/1089-134691-0005.flac'.

- **Evaluation metric.** As an ASR error correction benchmark, HP employs WER as the primary metric to evaluate the system performance. Nevertheless, prior work [2] has pointed out that WER can be too coarse-grained for describing the performance of ASR models. Furthermore, [82] raise community awareness regarding the problems caused by the optimistic bias toward ASR accuracy. In the future, we aim to provide more annotations for spoken language, e.g., entity spans and dependency structure. Accordingly, a comprehensive evaluation framework can be established to assess the quality and interpretability of output from the LLM-enhanced ASR system.

- **Robustness in reality.** HP benchmark covers mainstream domains where ASR tasks are usually deployed. However, as shown in [57], no single validation or test set from public datasets is sufficient to measure transfer to real-world audio data. Since all test sets of HP benchmark are drawn from existing ASR corpus, despite enhancing the WER performance, we are unable to ascertain the extent to which it can mitigate the gap between well-trained ASR models and real-world application scenarios. Furthermore, considering the discrepancy between spoken language and written language, more efforts are required from both speech and NLP communities to build a human-like robust ASR system beyond single modality [18].

## Broader Impact

With recent advances in using large-scale neural language models to solve problems once believed to be challenging for machines to learn and understand, we believe it is timely to move to the next

milestone: providing publicly accessible n-best hypotheses as transcription resources from LLM decoding. This motivation inspires this work, offering a collection of hypotheses paradise, inspired by in-context learning.

- *Who may benefit from this research:* Researchers working on speech technology and language model based error correction; as well as the users using the related techniques for responsible and reproducible machine learning technology.

- *Who may be put at disadvantage from this research:* When our work revealed that open-source hypotheses can be used to generate malicious recognition, we understood the responsibility of properly explaining the results to the public and providing reproducible evaluations. We have discussed terms of use for reusing these hypotheses with legal and regulatory experts, addressing potential risks and concerns.

- *Whether the task/method leverages biases in the data:* To alleviate possible bias in the data and model, we have made efforts to design reproducible metrics and to evaluate a wide variety of reproducible data sources and training configurations. We have also conducted user studies to highlight potential bias in "terms of use" provided in our Github repo.

## Maintenance Plan

- *Who will be supporting/hosting/maintaining the dataset?* Hypotheses Paradise has been actively maintained by the authors of this paper. We are still actively updating the dataset that focus on specific ASR scenario, which are noise-robust ASR and multi-lingual ASR. In INTERSPEECH 2023, we will have a tutorial to introduce the related Hypotheses Paradise-V2 with some excited experimental results. Furthermore, we also open the link to collect more hypothesis-transcription pairs from public.

- *How can the owner/curator/manager of the dataset be contacted?* To contact the main developers, we encourage users to use our emails: {chen1436,yuchen005}@e.ntu.edu.sg, huckiyang@gatech.edu.

- *Is there an erratum?* Users can use GitHub to report issues/bugs, and we would actively improve the codes accordingly. We also have a HuggingFace Model card under an non-profit organization in https://huggingface.co/datasets/PeacefulData/HP-v0.

- *Will the dataset be updated?* Yes, we are actively updating Hypotheses Paradise codes and data sources. Users could get information and the newly updated version through our GitHub repository.

- *If the dataset relates to people, are there applicable limits on the retention of the data associated with the instances?* No for the Dataset.

- *Will older versions of the dataset continue to be supported/hosted/maintained?* Yes, we will keep the old version that generated by Whisper. All versions can be found on our GitHub repository

- *If others want to extend/augment/build on/contribute to the dataset, is there a mechanism for them to do so?* We maintain Hypotheses Paradise on GitHub and we encourage all users to share their ideas to extend Hypotheses Paradise to more speech recognition cases. Users can use GitHub to report issues/bugs, and send us emails to discuss solutions.

## Acknowledgement

This research is supported by the National Research Foundation, Singapore under its AI Singapore Programme (AISG Award No: AISG2-100E-2022-102). The computational work for this article was partially performed on resources of the National Supercomputing Centre, Singapore (https://www.nscc.sg). The corresponding author is supported by Wallace H. Coulter Fellowship.

