# Hypotheses Paradise: An Open and Strong Baseline for Speech Recognition with Large Language Models

Chen Chen[1,†]    Yuchen Hu[1,†]    Chao-Han Huck Yang[2*]    Sabato Macro Siniscalchi[2,3]
Pin-Yu Chen[4]    Eng Siong Chng[1]

[1]Nanyang Technological University, Singapore; [2]Georgia Institute of Technology, USA
[3]Norwegian University of Science and Technology, Norway; [4]IBM Research AI, USA

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

(a) Predicted transcription ⇧ Correction Model ⇧ *Prompt* + N-best hypotheses list

(b) Hidden ⬈ ✚ ⬊ Pre-trained Weights | B=0 / A= $\mathcal{N}(0, \sigma^2)$ ⬊ ⬈ Input

Figure 3: (a) Structure of H2T-*ft*. (b) Reparametrization in H2T-*LoRA*. Solid box denotes the module is fixed during tuning while dashed box stands for trainable. Blue color denotes the weights has been pre-trained on another dataset.

# 4 ASR Error Correction from Hypotheses to Transcription

We hereby introduce a hypotheses-to-transcription (H2T) training scheme utilizing the collected HP dataset to enhance ASR performance with LLM integration. With limited labeled data, in-context learning is employed to form task-specific prompts and in-domain demonstrations: Linguistic knowledge in LLM is exploited without parameter tuning. Furthermore, we present two trainable methods fine-tuning (*ft*) and H2T-*LoRA* to learn the hypotheses-to-transcription mapping when a sufficient amount of labeled data is available.

## 4.1 Self-activated In-context Learning

We present a "self-activated" in-context learning mechanism in Fig.2, where a multi-turn prompting is generated to activate consistent knowledge in LLMs. This mechanism can be applied for both *zero-shot* and *few-shot* settings, where *few-shot* setting requires some in-domain hypotheses-transcription pairs (blue box in Fig.2). Under this setup, an LLM first explains the task it is working on, then generates an example from its understanding, and finally produces the actual task output. In our experiments, we noticed that LLMs can mimic an N-best hypotheses list consisting of utterances with similar pronunciation, showing that LLMs have perceived acoustic information during pre-training.

In *few-shot* learning, we provide some in-domain training examples as a demonstration for LLM input, before an LLM performs on the target task. In this case, we can also insert the domain information into the demonstration, as shown in Fig.2. Furthermore, we explore the effect of this domain-hint prompting on zero-shot setting with relative experiments in 5.5.

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

| AM | Baseline | H2T-*LoRA* | $o_{nb}$ | $o_{cp}$ |
|----|----------|------------|----------|----------|
| Whisper | 6.3 | $5.0_{-20.6\%}$ | 4.1 | 3.1 |

## Comparison with Error Correction Baselines

Compared with speech recognition, error correction techniques are more actively explored in the NLP community, as known as Grammatical Error Correction (GEC). However, directly evaluating well-trained GEC on our benchmark is unfair: the "Grammar" might be different between ASR transcription and normal text. To address it, we first train two error correction models (T5-large and FLAN-T5-large) in a typical sequence-to-sequence manner, which consumes the top-1 hypothesis to predict the true transcription. Furthermore, we also reproduce the MM-GEC from [22], where speech signal is utilized to provide grounded information with a cross-attention approach. Specifically, we employ top-1 hypothesis as ungrammatical text and recall speech from original dataset to compose paired data. For comparison, we select WSJ and LRS2 from HP dataset as representatives, since H2T-*LoRA* respectively achieves significant (51.1%) and moderate (12.9%) improvements in terms of WER. The experimental results are reported in Table 10.

Table 10: Comparison WER result with other GEC methods that are trained with top-1 hypothesis and transcription pairs. "$o_{nb}$" and "$o_{cp}$" respectively denote n-best oracle and compositional oracle that are defined in 5.2.

| Dataset | Baseline | T5 | FLAN-T5 | MM-GEC | H2T-*LoRA* | $o_{nb}$ | $o_{cp}$ |
|---------|----------|-----|---------|--------|------------|----------|----------|
| WSJ | 4.5 | 4.5 | 4.2 | 3.7 | $\mathbf{2.2}_{-51.1\%}$ | 4.1 | 1.2 |
| LRS2 | 10.1 | 9.9 | 9.8 | 10.1 | $\mathbf{8.8}_{-12.9\%}$ | 6.9 | 2.6 |

## Noise-robust ASR Results

Apart from CHiME-4, we add two noise-robust ASR datasets to verify the effectiveness of proposed H2T-*LoRA*. The first is NOIZEUS dataset [42], which is a test set containing 30 sentences corrupted by eight different real-world noises from Aurora-2 dataset [35] at different SNRs. Here we only select the 5dB SNR version for evaluation. To simulate noisy training data, we add these noises to LibriSpeech [69] *train-100* data at random SNRs of $\{0, 5, 10, 15, 20\}dB$. The second is VoiceBank-DEMAND dataset [89], which contains 11572 sentences in training set and 824 sentences in test set. The noisy training set contains 10 noises from DEMAND dataset [84]. To simulate more challenging mismatched train-test conditions, we add three different kind of noises [59] at 0dB to original clean test set. As shown in Table 11, our proposed H2T-*LoRA* presents significant effectiveness on noise-robust ASR task, with consistent improvement on different noise conditions.

---

[15]https://huggingface.co/ziqingyang/chinese-llama-2-7b

Table 11: WER (%) results of noise-robust ASR on NOIZEUS and VoiceBank-DEMAND datasets. "$o_{nb}$" and "$o_{cp}$" respectively denote n-best oracle and compositional oracle that are defined in 5.2.

| Test set | Noise | Baseline | H2T-*LoRA* | Oracle $o_{nb}$ | $o_{cp}$ |
|---|---|---|---|---|---|
| NOIZEUS | airport | 13.2 | $12.4_{-6.1\%}$ | 11.8 | 7.9 |
| | babble | 21.4 | $18.4_{-14.0\%}$ | 15.4 | 9.9 |
| | car | 19.9 | $18.0_{-9.5\%}$ | 11.7 | 7.0 |
| | exhibition | 17.7 | $17.6_{-0.6\%}$ | 12.3 | 9.9 |
| | restaurant | 15.6 | $12.9_{-17.3\%}$ | 10.8 | 8.3 |
| | station | 19.2 | $15.0_{-21.9\%}$ | 14.0 | 9.9 |
| | street | 21.4 | $20.1_{-6.1\%}$ | 13.8 | 11.2 |
| | train | 21.9 | $21.3_{-2.7\%}$ | 13.8 | 11.6 |
| VoiceBank-DEMAND | Helicopter | 10.3 | $6.8_{-34.0\%}$ | 5.9 | 4.0 |
| | Baby-cry | 10.6 | $6.5_{-38.7\%}$ | 5.7 | 3.5 |
| | Crowd-party | 26.5 | $20.6_{-22.3\%}$ | 17.8 | 12.2 |

## Code-switching ASR Results

Code-switching speech is defined as speech that more than one language within an utterance. Despite the remarkable success of ASR, code-switching speech recognition (CS-ASR) is still a challenging task due to grammatical structure complexity. We conduct experiment using our proposed H2T-*LoRA* on SEAME dataset [62], which is a conversational Mandarin-English CS corpus collected in Singapore. The training set includes 100 hours of CS speech from 134 speakers. Additionally, SEAME has two official test sets *SEAME-man* and *SEAME-sge*, each consisting of 10 speakers. which are respectively toward Mandarin and English speech. *Test-man* is biased towards Mandarin speech and *Test-sge* towards English.

Table 12: Mixed Mandarin-CER (%) and English-WER (%) results of CS-ASR on SEAME dataset.

| Test set | Metric | Baseline | H2T-*LoRA* |
|---|---|---|---|
| *SEAME-man* | CER | 16.6 | $12.9_{-22.3\%}$ |
| *SEAME-sge* | WER | 23.3 | $19.7_{-15.5\%}$ |

## Limitations

Though the proposed HP benchmark provides a new paradigm of generative error correction for ASR, we analyze and discuss the limitations of this work from the following perspective:

- **Evaluation metric.** As an ASR error correction benchmark, HP employs WER as the primary metric to evaluate the system performance. Nevertheless, prior work [2] has pointed out that WER can be too coarse-grained for describing the performance of ASR models. Furthermore, [83] raise community awareness regarding the problems caused by the optimistic bias toward ASR accuracy. In the future, we aim to provide more annotations for spoken language, e.g., entity spans and dependency structure. Accordingly, a comprehensive evaluation framework can be established to assess the quality and interpretability of output from the LLM-enhanced ASR system.

- **Robustness in reality.** HP benchmark covers mainstream domains where ASR tasks are usually deployed. However, as shown in [58], no single validation or test set from public datasets is sufficient to measure transfer to real-world audio data. Since all test sets of HP benchmark are drawn from existing ASR corpus, despite enhancing the WER performance, we are unable to ascertain the extent to which it can mitigate the gap between well-trained ASR models and real-world application scenarios. Furthermore, considering the discrepancy between spoken language and written language, more efforts are required from both speech

and NLP communities to build a human-like robust ASR system beyond single modality [16].

## Broader Impact

With recent advances in using large-scale neural language models to solve problems once believed to be challenging for machines to learn and understand, we believe it is timely to move to the next milestone: providing publicly accessible n-best hypotheses as transcription resources from LLM decoding. This motivation inspires this work, offering a collection of hypotheses paradise, inspired by in-context learning.

- *Who may benefit from this research:* Researchers working on speech technology and language model based error correction; as well as the users using the related techniques for responsible and reproducible machine learning technology.
- *Who may be put at disadvantage from this research:* When our work revealed that open-source hypotheses can be used to generate malicious recognition, we understood the responsibility of properly explaining the results to the public and providing reproducible evaluations. We have discussed terms of use for reusing these hypotheses with legal and regulatory experts, addressing potential risks and concerns.
- *Whether the task/method leverages biases in the data:* To alleviate possible bias in the data and model, we have made efforts to design reproducible metrics and to evaluate a wide variety of reproducible data sources and training configurations. We have also conducted user studies to highlight potential bias in "terms of use" provided in our Github repo.

## Maintenance Plan

- *Who will be supporting/hosting/maintaining the dataset?* Hypotheses Paradise has been actively maintained by the authors of this paper. We are still actively updating the dataset that focus on specific ASR scenario, which are noise-robust ASR and multi-lingual ASR. In INTERSPEECH 2023, we will have a tutorial to introduce the related Hypotheses Paradise-V2 with some excited experimental results. Furthermore, we also open the link to collect more hypothesis-transcription pairs from public.
- *How can the owner/curator/manager of the dataset be contacted?* To contact the main developers, we encourage users to use our emails: {chen1436,yuchen005}@e.ntu.edu.sg, huckiyang@gatech.edu
- *Is there an erratum?* Users can use GitHub to report issues/bugs, and we would actively improve the codes accordingly. We also have a HuggingFace Model card under an non-profit organization in https://huggingface.co/datasets/PeacefulData/HP-v0.
- *Will the dataset be updated?* Yes, we are actively updating Hypotheses Paradise codes and data sources. Users could get information and the newly updated version through our GitHub repository.
- *If the dataset relates to people, are there applicable limits on the retention of the data associated with the instances?* No for the Dataset.
- *Will older versions of the dataset continue to be supported/hosted/maintained?* Yes, we will keep the old version that generated by Whisper. All versions can be found on our GitHub repository
- *If others want to extend/augment/build on/contribute to the dataset, is there a mechanism for them to do so?* We maintain Hypotheses Paradise on GitHub and we encourage all users to share their ideas to extend Hypotheses Paradise to more speech recognition cases. Users can use GitHub to report issues/bugs, and send us emails to discuss solutions.