# OpenReview forum: "HyPoradise: An Open Baseline for Generative Speech Recognition with Large Language Models"
_NeurIPS.cc/2023/Track/Datasets_and_Benchmarks — NeurIPS 2023 Datasets and Benchmarks Poster_

### Official Review · Reviewer_7FEE · 2023-07-05
**A LLM-based ASR error correction method**

**Rating:** 6
**Confidence:** 4
**Correctness:** Yes
**Clarity:** Yes

**Strengths:**


1. The authors propose three approaches, including self-activated in-context learning and hypotheses-to-transcription training (H2T-ft and H2T-lora).

2. The dataset is constructed on multiple ASR corpora with different domains on 9 datasets.

3. Experimental results show that the proposed method can significantly reduce the WER of ASR.

**Additional Feedback:**

N/A

**Documentation:**

Yes

**Limitations:**


1. This paper is more like a regular paper, not a paper on Track Datasets and Benchmarks. Because this paper focuses on utilizing LLM to improve ASR error correction, and constructing the pairs of N-best hypotheses and accurate transcriptions is one of the step of the method.

2. The paper only compares the proposed method with LM reranking method. It would be better to compare to more ASR error correction approaches.

**Opportunities For Improvement:**


See Limitations.

**Relation To Prior Work:**


This task is also related to system combination, which re-generates the final results according to N-best results in machine translation and speech recognition tasks. But the authors do not mention them in related work.

**Summary And Contributions:**


This paper introduces a benchmark to utilize external large language models for ASR error correction, where utilize N-best hypothesis to re-generate the final prediction. To achieve this, the authors first conduct a "Hypotheses Prardise" (HP) dataset, which contains the pairs of N-best hypotheses and corresponding accurate transcription. Second, the proposed several ASR error correction techniques with LLM for zero-shot, few-shot, and finetuning settings. Results show the effectiveness of the proposed methods.

---

> ### Author Response · Authors · 2023-08-17
> **Response for Reviewer 7FEE**
>
> We sincerely appreciate Reviewer 7FEE for valuable comments. This paper is a preliminary attempt to establish an open-source and reproducible dataset for generative error correction. Now we response to your concerns.
> - ***Q1: Paper is more like a regular paper.***
> We do appreciate and understand the reviewer’s point of view. However, we would like to respectfully point out that our goal is to create proper benchmarks and baselines for the new research topic on “generative error correction” in ASR, which fits into the scope of the “Dasetsets and Bechmarks” track. The dataset challenge is that we could not find any public dataset covering a rich amount of hypothesis-transcription pairs for this task. Therefore, we decided to generate the HP dataset from common ASR domains using mainstream corpora  before proposing any technique. Since our curated dataset and the ASR evaluations with LLMs are our major contribtuions,   we believe our work is a good fit to the dataset and benchmark track in Neurips 2023. Furthermore, the proposed approaches aimed at demonstrating that the HP dataset contains rich information, which can be leveraged to significantly improve the quality of ASR output and cause a paradigm shift from the conventional rescoring approach in the ASR field. We firmly believe HP would be a unique and useful dataset that provides new insights into LLM-enhanced ASR.
> - ***Q2: Comparison with other ASR error correction approaches.***
> We thank the reviewer for the suggestion. We investigated other error correction baselines and reported the comparison results in Appendix Table 10. Furthermore, since we just incorporated AISHELL-1 (Mandarin corpus) into the HP dataset, more error correction baselines are being reproduced (Please kindly refer to the answer of Q1 for Reviewer LMrw).
> - ***Q3: System combination.***
> We would like to respectfully point out that our method employ single single ASR model for N-best list generation. WavLM and Whisper are deployed for different settings, but we did not combine the hypotheses of them. Furthermore, system combination in ASR usually refer to those methods that integrate multiple ASR systems, e.g., [1] combines 8 system to improve the WER performance.
>
> ***Reference***
> [1] K. J. Han, A. Chandrashekaran, J. Kim, and I. Lane, “The CAPIO 2017 conversational speech recognition system,” arXiv preprint
> arXiv:1801.00059, 2017.

---

> > ### Comment · Reviewer_7FEE · 2023-08-25
> >
> > Thank you for your detailed response, which helps me understand better.

---

### Official Review · Reviewer_4B4v · 2023-07-19
**Review: Hypotheses Paradise: An Open and Strong Baseline for Speech Recognition with Large Language Models**

**Rating:** 6
**Confidence:** 4
**Correctness:** The methods seem correct, and the ove…

**Strengths:**

A clear strength of the paper is the need for a consistent benchmark for ASR re-ranking. ASR hypothesis re-ranking is a common part of many production ASR systems, and thus, improvements in this field could have significant impact on the perceived quality of ASR systems. Most traditional ASR re-ranking/re-writing methods evaluate on a collection of ad-hoc benchmarks, generated by a wide range of root ASR models, which can make it difficult to directly compare the performance of the methods directly.

The proposed HP benchmark is fairly large, and contains both training and test sets, which means that non zero-shot methods (such as the H2T-ft and H2T-lora models) can be evaluated on the same data. Further, the HP benchmark covers a fairly large number of domains: (LibriSpeech, CHiME-4, WSJ, SwitchBoard, CommonVoice, TED-lium-3, LRS2, ATIS and CORAAL), with attention to the global nature of data (included several accents in CommonVoice, and the inclusion of CORAAL).

The introduced model baselines are interesting, and while somewhat limited, demonstrate that there is significant space for potential improvement over baseline approaches (even strong baselines such as Whisper).

**Additional Feedback:**

Overall, while the paper does solve a useful problem in the ASR benchmarking space, I am concerned that the paper focuses too much on the introduced baselines over the selection of the benchmark, and releasing the tools and data carefully. The paper could be significantly improved with additional discussion of the motivations and choices made when creating the dataset and benchmark, and well as additional discussion of the choices of baseline methods and scoring approaches.

**Clarity:**

The paper is sufficiently clear, however could include additional detail (see opportunities for improvement, limitations, prior work)

**Documentation:**

There is some detail on dataset information, including a discussion of dataset maintenance and broader impact, however the documentation could be significantly improved:

**No inclusion of Dataset card**: It could be considered best practice to include a dataset card [2,3], or other similar formulaic discussion of the details of the dataset. While the included discussion of dataset maintenance is helpful, it does not discuss all of the limitations and details suggested by one of these methods.

**Some data missing from github repo**: The data for ARIS, CORAAL, LIUM-3, LRS-2 and Librispeech are missing from the github repository.

**HP Dataset Legal Agreement does not match with MIT License**: It seems like the legal agreement PDF and the MIT license are at odds - it's not clear exactly what is being used.

**Evaluation scripts are not provided**: One of the key components of a benchmark are scripts that can be used for evaluation, which ensure that results (such as WER) are computed in the same way. It would be good to include at least the WER evaluation (including text normalization) as part of the benchmark.

[1] Gebru, Timnit, et al. "Datasheets for datasets." Communications of the ACM 64.12 (2021): 86-92.
[2] Holland, Sarah, et al. "The dataset nutrition label." Data Protection and Privacy 12.12 (2020): 1.

**Ethics:**

No major ethical concerns.

**Limitations:**

This paper has a very limited discussion of limitations, and the limitaions are _entirely_ focused on the baseline models, and not the HP benchmark. For a paper which is introducing a new benchmark and dataset, I believe that such as discussion should be significantly more robust, and for the benchmark itself, a limitations section could include discussions of:
- The sampling process from each of the datasets, and the demographic/dataset balancing performed on the data (if any), and how this might impact the performance of the re-writing models/n-best models.
- The limitations of pulling data from existing data sources, instead of collecting novel utterances
- The limitations of focusing on english-only data compared to global applications
- The limitations (and benefits) of not producing a single output number, and how the dataset is balanced around the samples
- The limitations of fine-tuning the n-best models on Librispeech, instead of on each dataset specifically

For example, the authors write in the broader impacts section of the appendix: "When our work revealed that opensource hypotheses can be used to generate malicious recognition, we understood the responsibility of properly explaining the results to the public and providing reproducible evaluations. We have discussed terms of use for reusing these hypotheses with legal and regulatory experts, addressing potential risks and concerns," It would be nice to see some of the discussion, and have additional explanations of what was discussed, and what the potential risks and concerns might be.

Similarly, there is no discussion of the user studies of potential bias, that was mentioned in the broader impacts section.

For the models themselves, the limitations are also relatively weak, and do not discuss limitations of using LLMs for speech applications including:
- Introducing language-only hallucinations during the process (for generative models)
- Cost of LLMs (monetary, environmental, social)
- Discussion comparing re-writing to re-ranking, and understanding the differences between these models.

**Opportunities For Improvement:**

While the paper does have notable strengths, there are some significant limitations to the work:

**Discussion of Sampling and Collection Process**: The discussion of the data selection could be improved in section 3.2. While the core details of the data are introduced, there is little motivation for each dataset, and there is no discussion of why each dataset was selected for the benchmark. Many of the choices are somewhat opaque (for example, why were 51,758 samples selected from Common Voice?, why use CHiME-4 vs. CHiME-5?), and some of the choices raise questions about the focus of the data (for example, WER 0 samples are filtered in librispeech, but not from other datasets, futher such examples are often the majority of samples available in real-world systems). I strongly believe that section 4/5 could be shortened (and likely, 5.1 could be eliminated) to make room for further discussion of the selection of the data: a key part of the benchmark.

**Discussion of the ASR system**: In section 3.1, the discussion of the ASR selection process could be more detailed. One of the key contributions of the method is that it unified the ASR model in use between papers (so as to evaluate rescoring/re-writing directly). Thus, additionally motivating the choice of these models would be nice (particularly as no transducer model was used, even though transducer models are relatively common for online speech recognition). Further, the decisions here are relatively opaque: why use a beam size of 60, and then filter for top-5 probabilities (instead of just using a beam size of 5). Why were the two models combined to form the dataset? Are there differences in base model quality? Why fine-tune WavLM on Librispeech (but not the other datasets in the metric)? Does Whisper overlap with existing datasets?

**Benchmark Scoring**: This paper does not assign a single score to models, and instead, model results are reported independently on each dataset. While this may be a good decision, it would be nice to have a discussion of why this is the case. Further, while using WER is necessary (given that re-writing models produce an updated version of the text, and not a re-ranking of the samples), it would be interesting to understand this decision. Does using WER disadvantage discriminative models purely based on a language usage-context?

**LLM Benchmarks on top-1 predictions**: It would be interesting to see the performance of the in-context LLMs (or even the H2T-ft models) when using the top-1 prediction alone. It seems like in some cases, the models may be only performing sensible re-writes, instead of actually using the alternatives provided by the ASR model.

**Noisy Data**: While the introduction/abstract of the paper motivate re-scoring by suggesting that ASR models struggle when confronted with adverse conditions (such as background noise), the chosen datasets have limited applicability (only CHiME, and to a lesser extent, CommonVoice, reflect this decision). It would be good to understand why this is the case, and perhaps think about including additional specific noise-focused ASR datasets or measures, for example CTIMIT,  VCTK, NOIZEUS, etc.

**English Data**: All of the datasets discussed appear to be english-language datasets (I'm not familiar with CORAAL). It would be good to have a discussion on this choice.

**Relation To Prior Work:**

While the paper does discuss some of the related work, it seems like there is not much attention paid to the existing discriminative methods [1,2,3] for hypothesis re-ranking. Given that the oracle re-ranking results are relatively strong, I would expect to see at least one method in the benchmark section which is discriminative, something like the RescoreBERT paper below.

It's also not entirely clear that ASR error correction based on LLMs/LMs is a fully novel paradigm. Papers such as [4-15] have clearly demonstrated that using language models and additional statistical methods to re-write ASR output predictions can lead to performance improvements, thus, it may be good to rephrase this contribution.

[1] Xu, Liyan, et al. "Rescorebert: Discriminative speech recognition rescoring with bert." ICASSP 2022-2022 IEEE International Conference on Acoustics, Speech and Signal Processing (ICASSP). IEEE, 2022.

[2] Udagawa, Takuma, et al. "Effect and Analysis of Large-scale Language Model Rescoring on Competitive ASR Systems." Annual Conference of the International Speech Communication Association. 2022.

[3] Kuo, Chin-Hung, and Kuan-Yu Chen. "Correcting, Rescoring and Matching: An N-best List Selection Framework for Speech Recognition." 2022 Asia-Pacific Signal and Information Processing Association Annual Summit and Conference (APSIPA ASC). IEEE, 2022.

[4] Namazifar, Mahdi, Gokhan Tur, and Dilek Hakkani-Tür. "Warped language models for noise robust language understanding." 2021 IEEE spoken language technology workshop (SLT). IEEE, 2021.

[5] Namazifar, Mahdi et al. “Correcting Automated and Manual Speech Transcription Errors using Warped Language Models.” ArXiv abs/2103.14580 (2021): n. pag.

[6] Hazen, Timothy J.. “Automatic alignment and error correction of human generated transcripts for long speech recordings.” Interspeech (2006).

[7] Wang, Xiaofei et al. “Exploring Methods for the Automatic Detection of Errors in Manual Transcription.” Interspeech (2019).

[8] Yang, Jinyi et al. “Towards Automatic Methods to Detect Errors in Transcriptions of Speech Recordings.” ICASSP 2019 - 2019 IEEE International Conference on Acoustics, Speech and Signal Processing (ICASSP) (2019): 3747-3751.

[9] Zheng, Da et al. “Directed automatic speech transcription error correction using bidirectional LSTM.” 2016 10th International Symposium on Chinese Spoken Language Processing (ISCSLP) (2016): 1-5.

[10] Weng, Yue et al. “Joint Contextual Modeling for ASR Correction and Language Understanding.” ICASSP 2020 - 2020 IEEE International Conference on Acoustics, Speech and Signal Processing (ICASSP) (2020): 6349-6353.

[11] Jung, Sangkeun et al. “Speech recognition error correction using maximum entropy language model.” Interspeech (2004).

[12] Chen, Zheng et al. “Pre-Training for Query Rewriting in a Spoken Language Understanding System.” ICASSP 2020 - 2020 IEEE International Conference on Acoustics, Speech and Signal Processing (ICASSP) (2020): 7969-7973.

[13] Mangu, Lidia et al. “Finding consensus in speech recognition: word error minimization and other applications of confusion networks.” Comput. Speech Lang. 14 (2000): 373-400.

[14] Song, Yuanfeng et al. “L2RS: A Learning-to-Rescore Mechanism for Automatic Speech Recognition.” ArXiv abs/1910.11496 (2019): n. pag.

[15] Dong, Linhao et al. “Speech-Transformer: A No-Recurrence Sequence-to-Sequence Model for Speech Recognition.” 2018 IEEE International Conference on Acoustics, Speech and Signal Processing (ICASSP) (2018): 5884-5888.

**Summary And Contributions:**

This paper introduces "Hypothesis Paradise," a collection of N-best predictions on utterances drawn from several dataset sources (LibriSpeech, CHiME-4, WSJ, SwitchBoard, CommonVoice, TED-lium-3, LRS2, ATIS and CORAAL), along with their ground truth asr transcription, designed for the evaluation of ASR-rescoring models. The paper also introduces several baseline models including zero-shot/few-shot LLM-based models based on in-context learning, and fine-tuning models (standard fine-tuning, LoRA) which adapt the models on the training dataset.

The claimed contributions are:
- HP dataset (316K total samples)
- Novel ASR error correction techniques
- New paradigm for ASR error correction based on LLMs/LMs

---

> ### Author Response · Authors · 2023-08-17
> **Response for Reviewer 4B4v (Q1~Q5)**
>
> We sincerely appreciate Reviewer 4B4v for giving a series of useful and valuable comments. We now try to interpret the misunderstanding and add more content that is inspired by your suggestions.
> - ***Q1:Discussion of sampling and collection process.***
> We apologize for the misunderstanding we may have caused. For the dataset selection, we mainly consider the domain characteristics. We want to cover the most common speech domains in ASR applications. This motivations might answer your questions: 1) For Common Voice, we select those samples with specific accent labels, then view it as the domain of accent English ASR. 2) For ChiME, the update of CHiME-5 compared CHiME-4 is mainly focus on speech separation task (https://spandh.dcs.shef.ac.uk//chime_challenge/CHiME5/data.html), and we find ChiME-4 is widely reported in noise-robust ASR works [1-5]. 3) For LibriSpeech, we mainly consider its popularity, although the low WER (1.8) has limited room for error correction. We observed that in most training examples (more than 240k in total 280k), the first hypothesis is correct. This imbalanced distribution would hinder LLMs from considering other hypotheses. This is the reason we filtered some samples with WER of 0, and remaining 40k samples of them to learn a better correction strategy.
> - ***Q2: Discussion of the ASR system.***
> For the 2 ASR system selection, we mainly consider 2 key conditions for ASR deployment: (i) a well-trained ASR but encountering domain mismatch and (ii) a universal ASR but lacking domain specificity. We explain the motivation behind our choices: 1) WavLM, 2) Whisper, and 3) Not using Transduces.
> (1) We select WavLM to address the following question: Given a well-trained ASR system, how can we improve its performance when we deploy it in new scenarios with domain mismatch? Therefore, in Table 3 we employ the WavLM as the hypothesis generator, and report the zero-shot and few-shot results. We choose WavLM mainly because it integrates the remarkable achievements of self supervised learning and ASR backbones in recent years, which usually is viewed as a common choice when we aim to set up a powerful ASR system. In practice, we have investigated several types of similar pipelines, i.e., SSL-Encoder-Decoder, and WavLM achieves the best WER performance on LibriSpeech, which serves as the most common benchmark for this topic.
> (2) We select Whisper as it usually serves as a universal ASR solution to fit various practical scenarios, where domain mismatch is no longer serious due to its large-scale training data. In this case, we want to demonstrate LLMs-based error correction can further enhance the WER performance by a large margin for each domain, as shown in Table 2. We replace Whisper with WavLM to provide LibriSpeech baseline in Table 2, as it serves as a better in-domain generator, thus evaluating if H2T method can further improve the performance. Notably, we also conduct some fine-tuning experiments with Whisper but can not achieve comparable performance with H2T methods. For example, on the ATIS dataset, finetuning a Whisper can improve WER from 8.3% to 3.9%, but H2T-LoRA achieves only 1.9%.
> (3) We conducted some experiments with Transducer-based ASR, but as you properly mentioned, it is difficult for LLMs-based correction methods to fit the requirement of system latency as an on-line decoding method. Therefore, we have decided to remove the relative experiments in the submitted version of our paper. Additionally, due to the higher WER of transducer method, the performance gain is significant and we are willing to attach the related results if you are interested.
>  - ***Q3: Beam size setting.***
> The beam size was set to 60 because that is the default parameter in ESPNet (https://github.com/espnet/espnet/tree/master/egs2/librispeech). More importantly, **setting the beam size equal to 60 and then filtering to 5 is totally different from directly setting a beam size to 5**. The former approach can produce hypotheses with higher quality due to larger search space. Furthermore, for some short utterances, it is unnecessary to generate many hypotheses.
> - ***Q4: Benchmark rescoring.***
> (1) In H2T-LoRA, we aim to train low-rank adapters as domain expert for each dataset, and they share the same foundation model. It is a typical approach when utilizing a fixed model and trainable sub-modules to various domains [6]. In the future, we aim to train a larger unified model to see the cross-domain performance. (2) As an ASR error correction benchmark, we should keep consistent with other ASR methods for comparison, where WER is viewed as primary metrics. We add the discussion of metric in Appendix (Limitations) .
> - ***Q5: Top-1 predictions.***
> Thanks for your suggestion. We have added some results with the top-1 hypothesis in Appendix Table 10, where T5 and FLAN-T5 are both employed as correction models. The comparison also demonstrate the importance to use n-best list.

---

> > ### Author Response · Authors · 2023-08-17
> > **Response for Reviewer 4B4v (Q6~Q11)**
> >
> > - ***Q6: Noisy data.***
> > Thanks for your comment. Currently, we focus on extending our method on noise-robust ASR. Since different noises pose a specific distribution on the speech signal, we believe that such a shift will also be reflected in the token distribution in the N-best list, which is easy for LLMs to perceive and correct. In response to your concern, we provide some noise-robust baseline in Appendix Table 11 to demonstrate the effect of our method, but leave more exploration in future work.
> > - ***Q7: Only include the English dataset.***
> > We appreciate your comment, which allows us to point out that we are currently working on expanding HP in order to cover other languages. Indeed, a new HP version will be released in the near future. However, to provide a preliminary yet meaningful response to your concern, we have added an experiment on a subset ( (16.7%) of the AISHELL-1 dataset [7] (Mandarin) and reported the related experimental results in Appendix Table 9. Furthermore, we present some results of the code-switching task on the multi-lingual SEAME dataset [8] in Appendix Table 12, where each spoken utterance contains Mandarin-English code-switching.
> > - ***Q8: Limitation.***
> > We thanks for your comments. For the LLM limitations you mentioned, we explain that: (1) In our H2T methods, all generative tokens are included in the input N-best list as we constrain the decoding space. (2) We understand building a large language model is resource-consuming. But in our case, we did not train any language model from scratch, and we utilize pre-trained models for all H2T learning. For ICL experiments, we use the API from OpenAI. (3) We attach a re-ranking baseline and analyze the upper bound referring to $o_{nb}$ . For more limitations of HP dataset, we would attach them according to your comments as soon as completing related experiments.
> > - ***Q9: Dataset card.***
> > Please see the website of Huggingface (https://huggingface.co/datasets/PeacefulData/HP-v0/viewer/default/train)
> > - ***Q10: Evaluation scripts .***
> > Thanks for your advice. Currently we use the sclite toolkit with text normalization to calculate WER, and the link is in Appendix "H2T Configuration".
> > - ***Q11:  HP Dataset Legal Agreement does not match with MIT License.***
> > The dataset is released under MIT license. There is no legal agreement on the dataset. The agreement of the fine-tuning code usage is only for avoiding the harmful usage case on the code itself instead of the released dataset. Following the recent concerns on the fine-tuning model release, we consult a legal team and get to only collect the users information requesting the fine-tuning code on the Legal agreement. Instead, the LLM usage is highly recommended by both NeurIPS review and the ethical review team internally. The authors have confirmed the Legal Agreement on the Code are not conflicts after consulting with a professional legal team before the paper submission.
> >
> > ***Reference***
> > [1] T. Menne, J. Heymann, A. Alexandridis, K. Irie, A. Zeyer, M. Kitza, P. Golik, I. Kulikov, L. Drude, R. Schluter, and et al., “The RWTH/UPB/FORTH system combination for the 4th chime challenge evaluation ,” in Proc. of CHiME-4 Workshop, 2016, pp. 49–51.
> > [2] J. Du, Y.-H. Tu, L. Sun, F. Ma, H.-K. Wang, J. Pan, C. Liu, J.-D. Chen, and C.-H. Lee, “The ustc-iflytek system for chime-4 challenge,” Proc. CHiME, vol. 4, pp. 36–38, 2016.
> > [3] Z.-Q. Wang, P. Wang, and D. Wang, “Complex spectral mapping for single-and multi-channel speech enhancement and robust ASR,” IEEE/ACM TASLP, vol. 28, pp. 1778–1787, 2020.
> > [4] H. Wang, Y. Qian, X. Wang, Y. Wang, C. Wang, S. Liu, T. Yoshioka, J. Li, and D. Wang, “Improving noise robustness of contrastive speech representation learning with speech reconstruction,” in ICASSP 2022.
> > [5] X. Chang, T. Maekaku, Y. Fujita, and S. Watanabe, “End-to-end integration of speech recognition, speech enhancement, and self-supervised learning representation,” arXiv preprint arXiv:2204.00540, 2022.
> > [6] Audhkhasi, Kartik, et al. "Modular Conformer Training for Flexible End-to-End ASR." ICASSP, 2023.
> > [7] Hui Bu, Jiayu Du, Xingyu Na, Bengu Wu, and Hao Zheng. Aishell-1: An open-source mandarin speech corpus and a speech recognition baseline, in O-COCOSDA, 2017.
> > [8] Dau-Cheng Lyu, Tien-Ping Tan, Eng Siong Chng, and Haizhou Li. Seame: a mandarin- english code-switching speech corpus in south-east asia, in ACISCA, 2020.

---

> > > ### Author Response · Authors · 2023-08-19
> > > **Response for Reviewer 4B4v**
> > >
> > > Thanks for your comments about the discussion of the limitations. We added a section in Appendix (Limitations) in the updated version.

---

> > > > ### Comment · Reviewer_4B4v · 2023-08-21
> > > > **Thanks for the response**
> > > >
> > > > Thanks for the detailed responses. Given the answers to my questions, I am relatively satisfied, and I hope that the authors choose to include the promised details/modification in the main paper.

---

> > > > > ### Author Response · Authors · 2023-08-22
> > > > >
> > > > > We are delighted to learn that our response has address the reviewer’s concerns. We will update the paper as suggested and we appreciate the reviewer for increasing the review score!

---

### Official Review · Reviewer_y46i · 2023-07-25
**review results (milestone!)**

**Rating:** 7
**Confidence:** 5
**Clarity:** very clear

**Strengths:**

This is timely important work for the community of ASR, showing how to integrate the LLMs (and PLMs) with acoustic models. This paper can be an ASR database/benchmark milestone.

The novel dataset they introduced, "Hypotheses Paradise," is used to train and test this integrated system. The authors also proposed an in-context learning method to finetune with this dataset on LLMs.

The proposed system significantly reduces the word error rate (WER), indicating better performance than traditional re-ranking-based methods. As the system even corrects tokens missing in the N-best list.

**Additional Feedback:**

no additional problem

**Correctness:**

mostly correct!
but T5 is LLM? maybe using the term PLM is more precise.

**Documentation:**

sufficient

**Ethics:**

no problem

**Limitations:**

similar as above reasons

**Opportunities For Improvement:**

A pity is that this paper only limits the N-best rescoring. If the authors could show the structured data input (e.g., lattice), their work would be more substantial.

Of course, the acoustic models this paper used can't generate lattice. This is another problem.

**Relation To Prior Work:**

clearly discussed previous work

**Summary And Contributions:**

The proposed system uses LLMs or PLMs for error correction in the output transcriptions generated by the ASR system. It does this by providing the LLMs with the N-best decoding hypotheses from the ASR system—a typical rescoring work.

---

> ### Author Response · Authors · 2023-08-17
> **Response for Reviewer y46i**
>
> We sincerely appreciate Reviewer y46i for considering that this work is a milestone benchmark for the community of ASR. Your advice for improvement is professional and inspired us to add more acoustic confidence in the dataset.
> - ***Q1: Only limits the N-best rescoring. It’s better to show the structured data input.***
> Thanks for your suggestion and understanding. Speech lattices contain indeed richer information for correction. As the reviewer pointed out, even though we generate lattice, LLMs can not directly understand lattice as directly as text, and therefore we did not include lattice in the HP dataset. Nevertheless, we have collected scores from the acoustic model and will release them in the new version, which can be viewed as the confidence for the hypothesis evaluated by AM.
> - ***Q2: T5 is not LLM.***
> Many thanks for your reminder, and we have modified the title of Section 5.1.

---

### Official Review · Reviewer_Vw1k · 2023-08-02
**A very useful benchmark**

**Rating:** 8
**Confidence:** 5
**Clarity:** The paper is very well written.

**Strengths:**

First of all I'd like to state that the community has long awaited for such a paper to emerge. It's main strength is that it uses LLMs for correct ASR transcripts and evaluating if they do a good job at rescoring hypotheses. This benchmark is well needed because a variety of systems can be tested against it, including both the classical kaldi solutions and seeking how lattices can be embedded with LLM embeddings for rescoring scenarios, and also rule based rescoring injections. The proposed experimental setup with both an in-context learning scheme and a trainining scenario is fullfiling the variety of approaches currently used in academia and industry.

**Additional Feedback:**

I am delighted to see this work, however I am dissatisfied with the fact that a XXI century scientific paper discusses English only benchmark, basically treating English as a "default languages", we should not be doing this as a community, including multiple languages is not optional, it's a quality issue, we know that LLMs differ per language, we should be evaluating this. Especially a top conference like neurips should be rigid about this.

**Correctness:**

Of course there's always the question of "how robust is the benchmark if we changed the prompt". Multiple papers showed that LLM performance degrades absurdly if we make trivial changes in prompts (see https://aclanthology.org/2023.findings-acl.666/ for example). However this is hardly an argument against corectness. I find this work methodologically sound.

**Documentation:**

The repository could benefit from more documentation, a clearly defined reproducibility scheme, perhaps a DVC-like mechanism?

**Ethics:**

Ethically the paper is honky dory.

**Limitations:**

The authors discuss technical limitations and they are discussed sufficiently. They however do not discuss the limitations of doing an english only benchmark.

**Opportunities For Improvement:**

1. The benchmark needs operational improvements. Please provide a method for showcasing results (a markdown table in github at minimum) and a clear path for reporting results for new approaches that will be integrated into a leadearboard (ex. PRs with github to a markdown table). This needs to be a continous running benchmark not a one time shot.

2. Please consider evaluating more languages. It is 2023 and and the word "english" in your paper is present twice when describing data sets. English **is not a default language of the world**. At minimum per the Bender rule please write explicitly that you benchmark english language and also discuss the limitations of this approach. Preferably add other languages, if you add other languages I am willing to raise my evaluation of this paper, it is saddening how many approaches ignore the fact that english is not the only language around. There are data sets you can you for other languages, and we know prompting quality changes as languages change, having this in your benchmark would be very useful to the community.

3. Please consider adding the earnings21 data set as it provides a wide variety of ASR outputs and allows to compare ASR x LLM matrix https://github.com/revdotcom/speech-datasets/tree/main/earnings21 - which would make the results of the benchmark much more interesting, in general adding more ASRs would be a plus.

**Relation To Prior Work:**

1. What I would find a good complement to the paper is when authors considered citing vision papers that talked about many of the ASR-NLP problems in recent years. Writing vision papers is always risky and if we don't cite people for it, we're losing a culture of discussion. This is in no way a requirement, the paper is awesome as it is, but citing or at least just mentioning people who reported problems before you provided this benchmark is something that builds a community. See for example:

- https://aclanthology.org/2020.findings-emnlp.295/
- https://aclanthology.org/2023.findings-acl.495/
- https://aclanthology.org/2021.bppf-1.4/

I'm not saying these are all that are, but they are fairly cited works that inspired others, perhaps more can be found.


2. Please also consider citing works that benchmarked the asr/nlp border in other languages than english, many people strive to provide high quality asr-nlp solutions for their native tongues and most of them are not as priviliged as english/chinese speakers. Citing them allows such researchers to apply for grants and have funding for work on their languages. Example: https://ieeexplore.ieee.org/document/9854978 - but perhaps more can be found, WER for french is at 40% on spontaneous conversations, for spanish it's not much better. Many african languages also have these problems, it's nice to cite a variety of people trying to benchmark progress - there has been a benchmark for age related asr quality degradation, perhaps it's worth mentioning.

**Summary And Contributions:**

The authors propose an ASR error correction benchmark using LLMs. They use two schemes to evaluate the performance, an in-context and a finetuning scheme. As a result the authors provide a table of how well LLMs can recover from ASR errors.

---

> ### Author Response · Authors · 2023-08-17
> **Response for Reviewer Vw1k**
>
> We sincerely appreciate Reviewer Vw1k for considering that this work is novel and essential. It represents a preliminary attempt to establish an open-source and reproducible dataset. We also look forward to collaborating and interacting with the speech and generative error correction communities as part of our maintenance plan in future versions.
>
>
> - ***Q1: Should provide a method for showcasing results.***
> Thanks for your insightful comment. We have updated the Github and will provide continuous updates for this repo (https://github.com/Hypotheses-Paradise/Hypo2Trans).
>
> - ***Q2: Consider evaluating more languages not limited to English.***
> We appreciate your comment, which allows us to point out that we are currently working on expanding HP in order to cover other languages. Indeed, a new HP version will be released in the near future. However, to provide a preliminary yet meaningful response to your concern, we have added an experiment on a subset ( (16.7%) of the AISHELL-1 dataset [1] (Mandarin) and reported the related experimental results in Appendix Table 9. Furthermore, we present some results of the code-switching task on the multi-lingual SEAME [2] dataset in Appendix Table 12, where each spoken utterance contains Mandarin-English code-switching.
>
> - ***Q3: Consider adding an earnings21 dataset.***
> Thanks for your recommendation. The speech length of Earnings21 is too long, and we failed to find the utterance-level transcription from the given link. We would appreciate it if you could provide a reference about utterance-level transcription after splitting, and then we are willing to incorporate this dataset into the HP dataset.
>
> - ***Q4: Relation To Prior Work.***
> Your comments are thought-provoking. We have added more references according to your recommendation ([2][16][28][82][98], etc.), and discuss related limitations in the Appendix. Furthermore, we believe that the HP benchmark can serve as a bridge between ASR and NLP, and we are willing to continuously contribute to this community in the future.
>
> ***Reference***
> [1] Hui Bu, Jiayu Du, Xingyu Na, Bengu Wu, and Hao Zheng. Aishell-1: An open-source Mandarin speech corpus and a speech recognition baseline, in O-COCOSDA, 2017.
> [2] Dau-Cheng Lyu, Tien-Ping Tan, Eng Siong Chng, and Haizhou Li. Seame: a Mandarin-English code-switching speech corpus in south-east asia, in ACISCA, 2020.

---

### Official Review · Reviewer_LMrw · 2023-08-02
**Review for Hypotheses Paradise: An Open and Strong Baseline for Speech Recognition with Large Language Models**

**Rating:** 6
**Confidence:** 5
**Correctness:** Good
**Clarity:** /

**Strengths:**

1. Interesting idea. The author proposes an error reduction method for ASR and leverages the LLMs model for dataset creation, which is a new and interesting idea.
2. Comparison studies. The authors compare different LLMs and ASR models in various datasets and provide H2T-ft and H2T-LoRA for ASR error correction from hypotheses to transcription.
3. Evaluation. The authors present in-context learning results with interesting findings and demonstrate the results with distinct error reduction.

**Additional Feedback:**

/

**Documentation:**

Yes, the authors provide the data collection and organization, availability, and maintenance details. But they should elaborate more on ethical and responsible use.

**Ethics:**

/

**Limitations:**

It includes English ASR reduction and could be extended to multi-lingual scenarios.

**Opportunities For Improvement:**

1. Compare with the current error reduction method. It is recommended to compare the model with a general ASR improvement method such as FastCorrect.
2. For ft and Lora, it could be better to provide a comparison with different LLMs (T5 and LLaMA).


**Relation To Prior Work:**

This work relates to previous work on ASR correction and LLMs finetuning.


**Summary And Contributions:**

The authors utilize external large language models (LLMs) for ASR error correction, where N-best decoding hypotheses provide informative elements for true transcription prediction, and propose a novel dataset, “Hypotheses Paradise” (HP).

---

> ### Author Response · Authors · 2023-08-17
> **Response for Reviewer LMrw**
>
> We sincerely appreciate Reviewer LMrw for considering that this work is interesting and novel, and your comments for improvement is professional and constructive.
> - ***Q1: Should compare with current error correction models such as “FastCorrect”.***
>   We thank the reviewer for the suggestion. We have added related references ([50]~[52]) to Fast Correct works and cited them in the revised Section 2.1. It should be pointed out that those works (e.g., FastCorrect, FastCorrect2) are all assessed on Mandarin data. We incorporate the Mandarin dataset AISHELL-1 in the HP dataset and report the H2T-LoRA in Appendix Table 9. However, FastCorrect pre-trained models are temporarily unavailable during the rebuttal period (see Github issue https://github.com/microsoft/NeuralSpeech/issues/115). We will keep trying to reproduce their results. Currently, we investigated other error correction baselines and reported the comparison results in Appendix Table 10.
> - ***Q2: Provide ft and LoRA comparison both on T5 and Llama.***
>   Thanks for the suggestion. We have updated Table 2 in Section 4.2, which respectively reports the 4 types of combinations of T5 / LlaMa and “ft” / “LoRA”. Furthermore, the results show that “LoRA” is a better strategy to learn H2T mapping, and LlaMa-LoRA achieves the best performance in most domains.
> - ***Q3: Extended to multi-lingual scenarios.***
>   We appreciate your comment, which allows us to point out that we are currently working on expanding HP in order to cover other
>  languages. Indeed, a new HP version will be released in the near future. However, to provide a preliminary yet meaningful response to your concern, we have added an experiment on a subset (16.7%) of the AISHELL-1 dataset [1] (Mandarin) and reported the related experimental results in Appendix Table 9. Furthermore, we present some results of the code-switching task on the multi-lingual SEAME dataset [2] in Appendix Table 12, where each spoken utterance contains Mandarin-English code-switching.
>
> **Reference**
> [1] Hui Bu, Jiayu Du, Xingyu Na, Bengu Wu, and Hao Zheng. Aishell-1: An open-source mandarin speech corpus and a speech recognition baseline, in *O-COCOSDA*,  2017.
> [2] Dau-Cheng Lyu, Tien-Ping Tan, Eng Siong Chng, and Haizhou Li. Seame: a mandarin-
> english code-switching speech corpus in south-east asia, in *ACISCA*, 2020.

---

### Author Response · Authors · 2023-08-17
**General Response for All Reviewers**

We sincerely thank the efforts of all reviewers. The comments are valuable, professional, and constructive for this work. To address your concern, our major modifications are summarized as follows:

- We expand HP with a Mandarin dataset from AISHELL-1, and demonstrate the effect of H2T-LoRA methods (Table 9) with a subset of the training set and a Chinese LlaMa.
- We add some error correction baselines (Table 10) trained with top-1 hypothesis and transcription pairs.
- We add an acoustic score for each hypothesis in HP dataset, which shows the confidence from the ASR acoustic model.
- We report more results on noise-robust ASR (Table 11) and code-switching ASR (Table 12).
- We expand the experiments for H2T-ft and H2T-LoRA (Table 2), where T5 and LlaMa are both tuned to learn H2T mapping with these two approaches.
- We update the Github repo, Huggingface (https://huggingface.co/datasets/PeacefulData/HP-v0), and references.

The modified content in paper is highlighted in blue. More details are explained in the comment for each reviewer.

---

### Decision · Program_Chairs · 2023-09-22

**Decision:**

Accept (Poster)

**Comment:**

The paper proposes an error correction benchmark for ASR (Hypotheses Paradise) and new baselines set with the use of LLMs (T2H-ft and T2H-Lora), demonstrating how well LLMs perform on the error correction task.

In the speech community we test error correction methods or new methods on rescoring hypothesis by varying the later component but selecting some ASR system, which is not standardized in the community, thus hard to compare between different works. Proposed benchmark will allow researchers to focus only on error correction methods, including investigation of how LLMs can perform on this task / can be used and integrated, to have fair comparison in future. However, in future the benchmark should be updated based on the more advanced general ASR development as this will change the type of error we need to correct.

All Reviewers are positive about the dataset and benchmarking of LLMs. They raised several important concerns which authors addressed mostly during rebuttal and included already in the updated version: e.g. extending to more languages (not only English), details on data collection and sampling, comparison with prior models on error correction, the beam size choice, improvement of documentation / github / dataset card / evaluation scripts, explaining issue with the license (related to the use of fine-tuning code, while dataset is under MIT), the choice of ASR systems used to generate dataset.

I agree with Reviewers that it is a "timely important work for the community of ASR, showing how to integrate the LLMs (and PLMs) with acoustic models." and moreover considered baselines (in-context learning scheme and a training scenario) are widely used methods in academia and industry.

I also would recommend authors to include all promised things (most are incorporated already), e.g. limitation section should be extended with discussion on the data collection itself, and possible issues that LLMs were trained on the ASR transcriptions in advance, so that future evaluations are done carefully with this.